# Discovery of an Endonuclease G-inhibitory Ku80-peptide protecting against leukemogenic rearrangements at the *MLL* breakpoint cluster

Julia Eberle[1,9], Ahmed Salem [1,9], Mara Hofmann[2,6,9], Anja Reisser[2,9], Yasser B. Ruiz-Blanco[3,9], Yasser Almeida-Hernandez [4], Boris Gole[1,7], Melanie Rall-Scharpf [1], Jessica Angulo-Capel [2,8], Thomas Monecke [5], Elsa Sanchez-Garcia [3,4] ✉, J. Christof M. Gebhardt [2,6] ✉ & Lisa Wiesmüller [1] ✉

Endonuclease G (EndoG) is an evolutionarily conserved enzyme that cleaves the *Mixed Lineage Leukemia* breakpoint cluster region (*MLL*bcr) under sub-lethal chemotherapeutic treatment conditions, causing leukemogenic chromosomal rearrangements. While endogenous inhibitors (EndoGI) control EndoG in lower organisms, no such EndoGI has been identified in mammalian cells. Due to the structural similarity of EndoGI from *Drosophila melanogaster* to the C-terminus (Ct) of human Ku80, we perform immunoprecipitation, surface plasmon resonance analysis and 3D molecular modeling, revealing binding of human EndoG to Ku80-Ct putatively between amino acid 110–184. Docking modeling predicts EndoGI-like peptides clustering around residues 686-707 of Ku80. Our experimental studies provide evidence that Ku80-Ct and 28-mer peptide Ku3 reduce *MLL*bcr breakage after doxorubicin treatment independently of DNA-PK activity. Proximity ligation and single molecule tracking studies show that Ku3 antagonizes Ku80-EndoG association and modulates chromatin-binding of EndoG. Such *MLL*bcr protection blocks EndoG´s pro-tumorigenic functions without limiting cytotoxicity, pursued for co-treatments that reduce secondary leukemia, a severe side effect of chemotherapy.

Patients with therapy-related acute myeloid leukemia (AML) or, more rarely, acute lymphoblastic leukemia (ALL) represent a subset of patients who have received cytotoxic therapies, including chemotherapy and/or radiotherapy for treatment of a prior malignancy. Therapy-related AML or ALL also is referred to as secondary leukemia and characterized by poor prognosis[1]. Chromosomal rearrangements in a 0.4 kb hotspot of clustered breaks, crossing the intron 11-exon 12 boundary in the *MLL* gene (*MLL*bcr), are the most frequently observed molecular changes in secondary leukemia and in de novo infant leukemia[1–3]. Compilation of breakpoint distribution data from patients together with experimental results have connected clustered breakage at the *MLL*bcr with sensitivity to various cytotoxic drugs, such as topoisomerase inhibitors. Additional cell context-dependent features destabilizing the *MLL*bcr range from chromosome loop anchoring by

---

CTCF and Cohesin, open chromatin (DNAse I hypersensitivity), formation of a complex non-B DNA secondary structure predisposing genomic regions to breakage, RNA Polymerase II binding, transcription and R-loop formation to replication stress as well as apoptotic cleavage[1,3–6].

Endonuclease G (EndoG) has been identified as a key molecule in these leukemia-causing processes[2,7,8]. This ancient nuclease was originally reported to reside in the intermembrane space of mitochondria, translocating to the nucleus at the onset of apoptosis, where it fragments the genome independently of caspases. However, apoptosis-unrelated functions during stress responses were suggested to exist before pro-apoptotic specialization during evolution, such as the involvement of EndoG in DNA recombination[9]. Meanwhile, it has become clear that genome cleavage by apoptotic nucleases can also proceed sublethally and, in turn, trigger cancer-causing genome alterations[2,8,10]. After replication stress, which can be induced by chemotherapy, sublethal breakage occurs. The breakage is followed by rearrangements at the *MLL*bcr. EndoG, which can process single- and double-stranded DNA as monomeric 3′-5′ exo- and dimeric endonuclease, respectively[11], was identified as a trigger for these processes, acting downstream of DNA damage tolerance and base excision repair (BER) components at the *MLL*bcr[6,7].

Therefore, inhibitors of EndoG are promising candidates to protect normal cells when DNA-damaging chemotherapeutics are used to kill cancer cells. So far, natural EndoG inhibitors have not been described in mammals. In *D. melanogaster*, EndoGI forms a complex with EndoG as demonstrated by the solved 3D crystallographic structure[12]. Loll and colleagues[12] reported that EndoGI Domain 1 from Drosophila is structurally most closely related to the C-terminal domain of Ku80 (Ku80-Ct, aa 566–732)[13]. Searching aa sequences related to Domain 1 reveals similarities with Ku C-terminal domain-containing proteins. Therefore, we wanted to test the hypothesis whether endogenous human EndoGI could be related to human Ku80-Ct and would protect from *MLL* rearrangements.

Ku80 is well-known to form a ring-shaped heterodimer with Ku70, which recognizes DNA ends with high affinity[14,15]. The Ku70/80 (Ku) heterodimer recruits the DNA-dependent protein kinase catalytic subunit (DNA-PKcs) to form the active holoenzyme DNA-PK at DNA double-strand breaks (DSBs), subsequently phosphorylating itself and downstream DNA damage response factors[16]. DNA-PK tethers DNA ends and assembles the XRCC4-DNA ligase IV (LIG4) ligation complex, executing non-homologous end joining (NHEJ), the major repair pathway for two-ended DSBs. Of note, Ku also has the capability to bind and translocate along DNA structures other than blunt DNA ends, namely, cruciform structures or regressed replication forks[17]. Forming the so-called tool belt around DNA, Ku also interacts with multiple DNA repair, chromatin and transcription factors other than the core NHEJ components[16,18]. It was therefore not surprising to see accumulating evidence for non-canonical Ku functions, namely in controlling BER during NHEJ to prevent formation of additional breaks[19], in regulating gene expression through interaction with the BER and redox factor Apurinic/Apyrimidinic Endonuclease 1 (APE1)/Redox Effector Factor-1 (REF-1)[20], in rRNA processing and hematopoiesis[21], in cytoplasmic DNA sensing, which potentiates T cell activation[22], or in promoting fork reversal and chemoresistance[23]. Along this line, Ku was classically viewed to protect DSBs from unscheduled exonucleolytic processing, but more recent data from Schizosaccharomyces pombe uncovered a role in fine-tuning end-resection and recombination-mediated replication restart upon binding to the single DNA end of reversed forks[24,25].

Here, we test the hypothesis that the Ku80-Ct resembles an endogenous, human EndoGI under conditions of treatment with the anthracycline doxorubicin, a clinically highly relevant genotoxic drug acting through DNA intercalation, adduct formation and generation of reactive oxygens[26–28]. Anthracyclines are widely used for the treatment of many tumors and have been associated with *MLL* rearrangements in

secondary leukemia[1]. We utilize pre-established *MLL*bcr reporter cells[29] and biochemical assays to monitor dynamic changes at the *MLL*bcr regarding EndoG binding, breakage and rearrangements[30]. Supporting our concept, we reveal that Ku80-Ct shows protective effects on the *MLL*bcr integrity with respect to all three outcomes, while opposing effects are noticeable for full-length Ku80. These activities do not dampen the cytotoxic effect of doxorubicin and can be separated from canonical Ku80 functions by mutant design and repair measurements of targeted DSBs. Using biomolecular simulations and experimental testing of in silico designed candidates, we identify peptides mimicking the effects of Ku80-Ct at the *MLL*bcr. We use proximity ligation assay, single molecule tracking and clustering analysis of the EndoG-Ku3 binding simulations to show that the top hit peptide interferes with EndoG-Ku80 complex formation and modulates chromatin binding of EndoG dimers. A shared motif of the candidate peptides plays an important role in this modulation. Based on these discoveries, we suggest that a peptide derived from Ku80-Ct can recapitulate the *MLL*bcr protecting effect of Ku80-Ct, highlighting the potential of peptide-based protective drugs as part of cytotoxic cancer treatment regimens to prevent secondary leukemias.

## Results

### Ku80-Ct mitigates *MLL*bcr rearrangements but not the repair component

We have previously shown that replication stress-inducing agents such as the chemotherapeutic drug doxorubicin induce EndoG binding to the *MLL*bcr, its cleavage and genomic rearrangements[6,7,30]. Aiming at the identification of a specific inhibitor of such tumorigenic processes, we sought to determine whether the C-terminal domain of Ku80, showing structural similarities to EndoGI from lower organisms[12], affects such therapy-induced rearrangements. Supporting our concept, we observed pull-down of EndoG when immunoprecipitating endogenous Ku80 (Supplementary Fig. 1a). In surface plasmon resonance measurements (SPR) with a Ku80 fragment containing the C-terminal amino acids (aa) 384–732, we observed a strong binding affinity to EndoG with an equilibrium dissociation constant $K_D$ of $72.7 \pm 13.2$ nM determined from single-cycle triplicate measurements (Fig. 1a, b). Therefore, we ectopically expressed the structurally defined C-terminal domain between aa 592 to 709 of Ku80 (Ku80-Ct) fused to the nuclear localization signal (NLS) (Fig. 1c). Importantly, this Ku80 fragment lacks the core of wild-type Ku80 (Ku80-wt), known to bind DNA ends and to form heterodimers with Ku70[31,32], as well as the major DNA-PKcs interaction site at the C-terminal end of Ku80-wt[13,33]. Additionally, we created the full-length Ku80 variant with aa exchanges E720A and E721A, selectively abrogating binding of DNA-PKcs[33]. To quantify *MLL*bcr rearrangements, we measured recombination at the *MLL*bcr in an *EGFP*-based reporter integrated into the genome of K562 leukemia cells following a 4 h treatment with doxorubicin (Fig. 1d)[29,30]. Ku80-wt and Ku80-Ct proteins were detectable in these reporter cells (Fig. 1e). Recombination measurements revealed similar frequencies in cells with and without ectopic expression of Ku80-wt. In contrast, the recombination frequency was halved after expression of Ku80-Ct (Fig. 1f). Recombination reduction by Ku80-Ct was neither accompanied by viability changes, estimated by SSC/FSC-gating during FACS analyses (Supplementary Fig. 1b), nor by changes in the cell cycle distribution or the percentage of apoptotic cells according to DNA content analysis (Supplementary Fig. 1c).

To understand whether Ku80-Ct affects the DNA exchange processes underlying *MLL*bcr rearrangements in doxorubicin-treated reporter cells[6], we performed DSB repair experiments using the same reporter construct but replacing the *MLL*bcr spacer between the mutated *EGFP* genes by a hygromycin resistance cassette that fails to stimulate recombination in response to genotoxic treatment[29] (Supplementary Fig. 2a). Using this construct, homologous DSB repair can be triggered by targeted cleavage upon expression of the

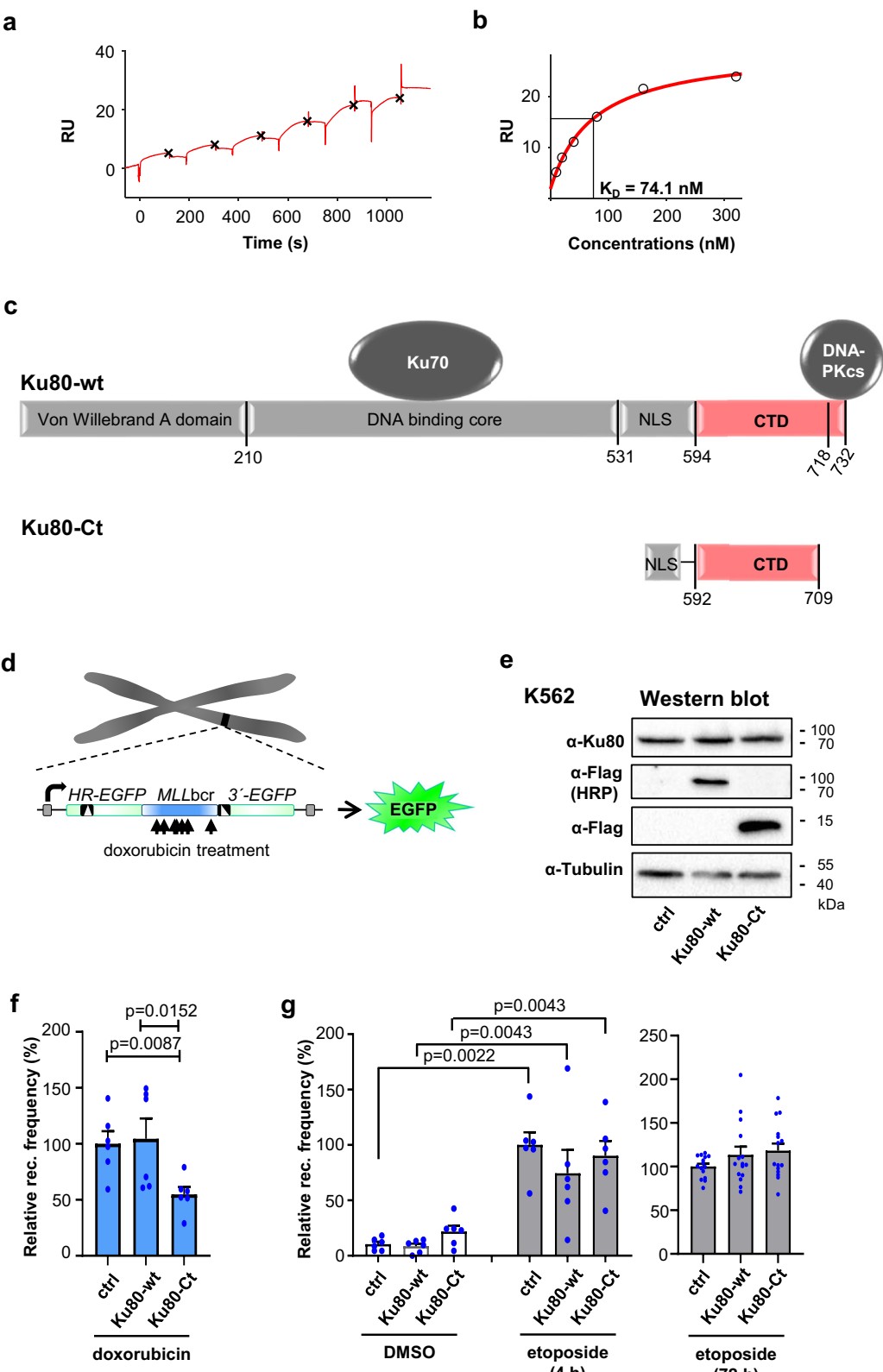

meganuclease I-SceI instead of chemotherapeutic treatment[34]. We found that in K562, neither co-expressed Ku80-wt nor Ku80-Ct altered DSB-induced recombination frequencies significantly when compared to the control (Supplementary Fig. 2b). To further check a possible effect on the balance between DSB repair pathways, we measured NHEJ using an *EGFP*-based reporter with I-SceI-recognition sites (Supplementary Fig. 2a, c). However, no Ku80-wt or Ku80-Ct dependent effect

on NHEJ was found (Supplementary Fig. 2c). To account for the fact that repair of DSBs generated by targeted cleavage might be different from repair of doxorubicin-induced breaks, we measured *MLL*bcr rearrangements in K562 reporter cells replacing doxorubicin by etoposide, also known to cleave the *MLL*bcr[7]. Though recombination frequencies were stimulated by etoposide- versus mock-treatment, Ku80-Ct had no effect on recombination (Fig. 1g) nor did it affect

**Fig. 1 | Ku80-Ct binds EndoG and mitigates doxorubicin-induced rearrangements adjacent to the *MLL*bcr. a, b** Surface plasmon resonance (SPR) experiment showing direct binding of a C-terminal Ku80 fragment (aa 384 – 732) to surface-coupled EndoG. **a** Sensorgram of a representative single-cycle measurement with EndoG as surface-coupled ligand and the C-terminal Ku80 fragment as analyte, applied in increasing concentrations (10, 20, 40, 80, 160, and 320 nM) (red line). The maximal response at equilibrium was marked for each concentration (black x). **b** Response-concentration plot from (a) shows the equilibrium response of each analyte concentration (black circles). A one-site binding fit (red line) was used to determine the dissociation constant $K_D$. The measurement was done in triplicate, resulting in a mean $K_D$ and standard deviation of 72.7 ± 13.2 nM. **c** Structure of Ku80 variants analyzed. Ku80-wt comprises the N-terminal Von Willebrand A domain[45], the central DNA end-binding core (aa 210–531)[32], which further encompasses aa 449–477 heterodimerizing with Ku70[31] and is followed by a nuclear localization signal (NLS)[86]. The structurally defined C-terminal domain (CTD, red) is predicted to form α-helices at aa 594–705 and an unstructured tail, which binds DNA-PKcs at the PIKK (PI3 kinase-like kinase) interaction motif (aa 718-732)[13,33]. For expression of Ku80-Ct, devoid of DNA, Ku70 and DNA-PKcs binding sites, a truncated CTD (aa 592-709) was fused to the NLS (aa 562–568). **d** Principle of *MLL*bcr rearrangement assay. Recombination measurements rely on quantification of EGFP-positivities post doxorubicin-treatment of K562 cells with chromosomally integrated reporter construct for recombination between differently mutated *EGFP* genes

encompassing the 0.4 kbp therapy-related *MLL*bcr[29,30]. **e** Ku80 variant expression in K562. Proteins were extracted 24 h post-electroporation of K562 cells with expression constructs for Ku80 variants specified in (a) or empty vector (ctrl). Western blot analysis was performed using antibodies for total Ku80 and the N-terminal DDK/Flag-tag of ectopically expressed variants (representative from 2 independent experiments). Uncropped Western blots in Source Data (uncropped images). **f, g** Recombination measurements post-chemotherapeutic treatment. K562 reporter cells, ectopically expressing Ku80 variants for 24 h, were treated with 2.0 µM doxorubicin (blue) for 4 h followed by 72 h drug-free culture, with solvent (DMSO, white) or 10 µM etoposide (gray) for 4 h followed by 72 h drug-free culture or for 72 h with etoposide and FACS analysis. Recombination (rec.) frequencies of EGFP-positive living cells in the population were normalized to the mean of ctrl values set to 100% per experiment (average for doxorubicin: $2 \times 10^{-5}$; for etoposide, 4 h: $6 \times 10^{-5}$ and 72 h: $5 \times 10^{-4}$) to calculate relative rec. frequencies. Data are presented as mean + SEM. Significances were calculated by the Kruskal-Wallis H-test followed by a two-tailed Mann-Whitney-U test and values of $p < 0.05$ indicated, gating strategies in Supplementary Fig. 1b, d. **f** Recombination measurements post doxorubicin treatment ($n = 6$ samples from 3 independent experiments). **g** Recombination measurements post etoposide treatment. DMSO and etoposide (4 h): $n = 6$ samples from 3 independent experiments. Etoposide (72 h): $n = 15$ samples from 5 independent experiments. Source data are provided as a Source Data file.

---

viabilities during etoposide treatment, regardless of the length of treatment (4 h vs. 72 h; Supplementary Fig. 1d). As compared to the mode-of-action of doxorubicin, which relies on DNA adduct formation and DNA intercalation and thereby induction of R-loops, replication stress and only after fork collapse DSBs, etoposide directly generates DSBs through topoisomerase II-poisoning[35]. We conclude that Ku80-Ct mitigates *MLL*bcr rearrangements, which cannot be explained by effects on DSB repair, cell cycle changes or cell viabilities.

## Ku80-Ct prevents accumulation of EndoG in chromatin and binding to the *MLL*bcr, protecting the integrity of the therapy-related breakpoint cluster

Rearrangements at fragile sites in the genome can be triggered by various stimuli such as secondary structures in the DNA, replication-transcription conflicts or DNA lesions, most notably breaks[1]. We have previously identified EndoG as the nuclease causing *MLL*bcr breakage during replication stress in different hematopoietic and tumor cell types[7,29]. Prior analysis had also shown that in response to doxorubicin treatment, EndoG forms a few but distinct nuclear foci and binds to the *MLL*bcr[30]. We therefore hypothesized that Ku80-Ct could protect the *MLL*bcr from such EndoG action. To test this hypothesis, we performed immunofluorescence microscopy on doxorubicin-treated HeLa cells, which have previously been shown to serve as a representative cell model for the processes leading to *MLL*bcr rearrangements[6,7,30]. Ectopic expression of Ku80-wt doubled the average number of nuclear EndoG foci in doxorubicin-treated cells (Fig. 2a, b), indicating enhanced chromatin association. Strikingly, Ku80-Ct abrogated foci formation (Fig. 2a, b). For comparison, mock-treated cells did not show statistically significant differences in EndoG foci numbers. We also performed microscopic analysis of nuclear Ku80 intensities under these conditions, which indicated lower nuclear Ku80 signals in cells ectopically expressing Ku80-wt both after doxorubicin- and mock-treatment (Supplementary Fig. 3a). To understand whether Ku80-Ct modulates binding of EndoG to the *MLL*bcr, we performed chromatin immunoprecipitation (ChIP) experiments post doxorubicin treatment. Consistent with EndoG foci analysis, expression of Ku80-wt caused the most pronounced binding of EndoG to the *MLL*bcr region as judged from PCR analyses of the EndoG pull-downs. Accordingly, PCR amplification of genomic DNA adjacent to the *MLL*bcr region was less prominent in the EndoG-specific precipitates after Ku80-Ct expression or in controls (Fig. 2c). Of note, this decrease was detectable even though PCR analysis of input genomic DNA revealed the highest integrity of

the *MLL*bcr region in cells expressing Ku80-Ct, showing 1.5-fold and 2.0-fold stronger *MLL*bcr-specific band intensities than in Ku80-wt expressing and control cells, respectively (Fig. 2d). These results suggested that Ku80-wt promotes recognition of the *MLL*bcr by EndoG, while Ku80-Ct protects this genomic region.

To more directly detect DNA damage at the *MLL*bcr, we also performed ChIP experiments using antibodies directed against histone H2AX phosphorylated on Ser139 (γH2AX), a well-established marker for various DNA lesions ranging from DSBs to sites of replication stress. As compared to control and Ku80-Ct-positive cells, cells ectopically expressing Ku80-wt showed enhanced PCR amplification of the *MLL*bcr region in γH2AX pull-downs (Fig. 2c). To understand whether such differences were related to overall changes in DNA repair, we evaluated γH2AX signals in the whole nucleus. As expected, doxorubicin treatment augmented the percentage of cells with elevated γH2AX signals, which was also seen in Ku80-Ct-positive cells (Supplementary Fig. 3b). We observed neither differences in the cell cycle distribution nor the percentage of apoptotic cells after Ku80-wt or Ku80-Ct expression, excluding related changes as possible confounding factors (Supplementary Fig. 3c). Even after prolonged (24 h) treatment with doxorubicin similar viabilities were found (Supplementary Fig. 3d). Collectively, our findings suggest that contrary to full-length Ku80-wt, which promotes *MLL*bcr association of EndoG in the chromatin and local DNA damage signaling without destabilizing the *MLL*bcr region, Ku80-Ct prevents chromatin association of EndoG and protects the integrity of the *MLL*bcr after treatment with the chemotherapeutic doxorubicin.

## Structure-based discovery of peptide inhibitors of human EndoG

We aimed to identify Ku80-based inhibitory peptides that target human EndoG in a similar way as the EndoG/EndoGI of Drosophila melanogaster[12] (Fig. 3a, top). First, we built a homology model of human EndoG (see the Methods section for details). Next, we generated 5050 Ku80-based peptides with a length of 20 or more residues, from the aa sequence of the C-terminal region of Ku80 (PDB ID: 1Q2Z[36]). We screened these peptides for their interaction with human EndoG, using the tool PPI-Detect[37], a sequence-based predictor of protein–protein and protein–peptide interaction likelihood. Peptide sequences accepted as interacting were those with a score higher than the score of the Ku80-Ct sequence. On this basis, we identified 356 peptide candidates with lengths between 20 and 60 residues. The

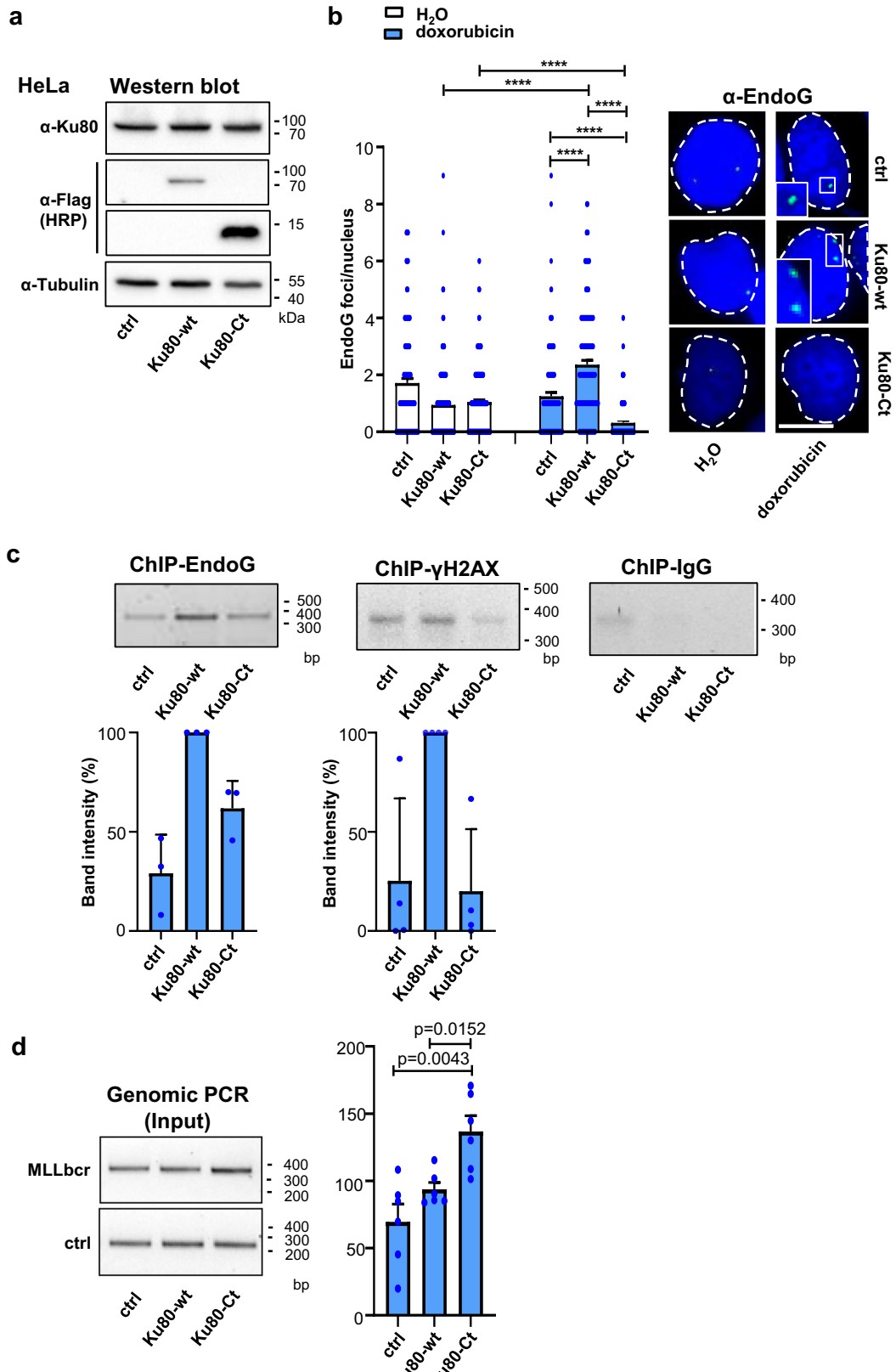

most conserved region among these peptides corresponds to residues 687-693 of Ku80 (sequence: TKEEASG) (Fig. 3a, bottom). We then carried out peptide–protein docking calculations on the human EndoG model using the subset of the 137 peptides with length ≤ 30 residues and the best interaction scores from PPI-Detect. We performed the docking calculations using CABS-Dock[38]. The docking analysis allowed

us to identify a binding site of the peptides on the protein corresponding to residues 172–252 of EndoG. To further evaluate and filter the peptides, we ranked them according to the extent of their interactions with EndoG. The selection criterion was that more than 50% of the peptide length should be in contact with EndoG, thus leading to 13 candidate peptides (Table 1 and Fig. 3b). The contacts were

**Fig. 2 | Ku80-Ct interferes with nuclear foci formation of EndoG, *MLL*bcr binding and breakage.** HeLa cells were transfected with expression constructs for Ku80-wt, Ku80-Ct or empty vector (ctrl) 24 h before experimental analysis. **a** Ku80 variant expression in HeLa. Protein extracts were subjected to immunoblotting to detect Ku80 and Flag-tagged variants (representative from 2 independent experiments). **b** Formation of EndoG foci in the nucleus. Immunofluorescence microscopy was performed after 4 h of treatment with $H_2O$ (white) or 0.5 μM doxorubicin (blue). Mean values + SEM are presented ($H_2O$, ctrl: $n = 160$; $H_2O$, Ku80-wt: $n = 230$; $H_2O$, Ku80-Ct $n = 250$; doxorubicin, ctrl: $n = 135$; doxorubicin, Ku80-wt: $n = 169$; doxorubicin, Ku80-Ct $n = 179$ nuclei from 2 independent experiments). Significances were calculated by the Kruskal-Wallis H-test followed by a two-tailed Mann-Whitney-U test and ****$p < 0.0001$ indicated. Insets display the highlighted regions at two-fold magnification. Scale bars indicate 10 μm. **c** Chromatin immunoprecipitation (ChIP) in doxorubicin-treated HeLa reporter cells. Sonified DNA bound by proteins was immunoprecipitated with antibodies specific for EndoG, γH2AX or incubated with control IgG. After DNA isolation, PCR amplifying a 0.3 kbp DNA fragment adjacent to the *MLL*bcr (EGFP/MLLbcr, MLLbcr-2) was performed,

band intensities evaluated, normalized to Ku80-wt (100%) and shown as mean + SD from 3 independent ChIP-EndoG and 4 independent ChIP-γH2AX experiments. Note that previous work already showed increased EndoG-binding to the *MLL*bcr in doxorubicin-treated cells[30]. Antibody specificities were high as visualized in Source Data (uncropped images), though it remains possible that additional low-abundance bands are present but not visible at the selected exposure. **d** Genomic PCR. Genomic DNA isolated from doxorubicin-treated HeLa reporter cells, such as during ChIP experiments resembling input DNA, was subjected to PCR analysis as in (**c**). *MLL*bcr band intensities were corrected by the intensities of the MLL control PCR (ctrl), amplifying a 0.2 kbp fragment 8 kbp downstream of the *MLL*bcr (Intron20 in the genomic *MLL*) and normalized to the mean of ctrl, Ku80-wt and Ku80-ct values (100%) each. Data from 6 independent experiments are presented as mean + SEM. Significances were calculated by the Kruskal-Wallis H-test followed by a two-tailed Mann-Whitney-U test and $p < 0.05$ indicated. PCR products were separated on SeaKem® LE Agarose (Lonza, Basel, Switzerland). Source data are provided as a Source Data file.

---

determined using a cutoff distance of 10 Å between the alpha carbon atoms of EndoG and the peptide. Two of these candidates correspond to the N-terminal part of Ku80-Ct, while the remaining eleven peptides belong to the C-terminal region. Noteworthy, the motif EEA is present in all the selected candidates at least once, which suggests that these residues may be critical for binding to EndoG. To further refine the list of peptide candidates, we calculated their water solubility with the Innovagen (https://pepcalc.com/peptide-solubility-calculator.php) and PROSO II[39] tools, resulting in four peptides predicted to be soluble in water (Table 1).

## Ku80-Ct-derived peptide Ku3 reduces *MLL*bcr rearrangements and breakage

Furthermore, we assessed whether the four EndoG inhibitory peptides identified in silico limit rearrangements at the *MLL*bcr. Two peptides (Ku2 and Ku4, Table 1) were derived from the N-terminal part of Ku80-Ct and another two from the C-terminal region (Ku1, Ku3) (Fig. 4a and Table 1). For recombination measurements, *MLL*bcr reporter cells derived from K562 were nucleofected with increasing amounts of each peptide, treated with doxorubicin to promote *MLL*bcr breakage[30] and thereafter released for rearrangements (Fig. 4b and Supplementary Fig. 4). Among the four peptides, we found that Ku3 downregulates recombination by ≤ 35% in a range from 50 to 175 μg peptide, while Ku1 showed such an effect only with 300 μg peptide (Fig. 4b and Supplementary Fig. 4a, b). Neither Ku2 nor Ku4 caused a recombination decrease, but elevated frequencies associated with data scattering at peptide levels of ≥ 300 μg (Supplementary Fig. 4c, d). Such aberrantly high frequencies were also noticed in water controls with Ku2 and Ku4 and accompanied by reduced viabilities (Supplementary Fig. 4c, d, lower panels), suggesting unspecific toxicities. Contrarily, after the introduction of Ku1 or Ku3, we did not observe viability changes after doxorubicin or mock-treatment (Supplementary Fig. 4a, b, lower panel). We confirmed that the scrambled version of Ku3 had no effect on recombination in the K562 reporter cells demonstrating sequence specificity (Supplementary Fig. 4e). Furthermore, we also measured decreases of the recombination frequencies without toxic effects when introducing Ku3 peptide at 25 to 75 μg into *MLL*bcr reporter cells derived from HeLa (Fig. 4c and Supplementary Fig. 4f). Efficient nuclear uptake of the peptide was visualized by use of chemically synthesized Ku3 N-terminally labeled with TAMRA dye (Fig. 4d). Collectively, peptides were found that inhibit *MLL*bcr rearrangements without mitigating the cytotoxic effects of the chemotherapeutic agent doxorubicin. Active peptides Ku3 and Ku1 were derived from the C-terminal extreme of Ku80-Ct, where the majority of peptide candidates clustered, and show a sequence overlap of 18 aa (ITK<u>EEA</u>SGSSVTA<u>EEA</u>KK) comprising the motif EEA twice, which was

present in all the identified candidates at least once (Fig. 4a and Table 1).

We next investigated the integrity of the *MLL*bcr region by genomic PCR analysis in the presence of peptide Ku3 (Fig. 5a). Doxorubicin treatment of the HeLa reporter cells caused a significant decrease (26%) of the *MLL*bcr-specific PCR band intensity (Fig. 5b). Nucleofection of the cells with 75 μg of Ku3 alleviated the doxorubicin-mediated destabilizing effect on the *MLL*bcr without inducing a change in mock-treated cells (Fig. 5b). Protection of the *MLL*bcr region in such chemotherapeutically treated cells was revealed by a 45% more pronounced PCR band intensity, matching the 1.5-fold increase in cells expressing Ku80-Ct as compared to Ku80-wt (Fig. 2d). Next, we performed PCR analysis in doxorubicin-treated K562 reporter cells, where Ku3 was found to more robustly downregulate *MLL*bcr rearrangements (Fig. 4b). To discriminate between different endogenous and reporter-based *MLL*bcr loci, we PCR amplified within the genomic *MLL*bcr locus (MLLbcr-1), adjacent to this genomic locus (MLLbcr-2) or across the boundary of the *HR-EGFP* cassette and the *MLL*bcr spacer within the recombination reporter (EGFP/MLLbcr-1), relative to stable control regions in the genome under conditions of chemotherapeutic treatment (*GAPDH*, *RARα*bcr, Intron20 in *MLL*) (Fig. 5a)[6,7,30]. On average, we observed a 1.5-fold increase of the *MLL*bcr band intensities (Fig. 5b, c and Supplementary Fig. 5a), which reflected endogenous and reporter-based *MLL*bcr integrities. We noticed less pronounced protection when monitoring the *MLL*bcr core (MLLbcr-1), likely due to stable secondary structure formation[1]. Importantly, we found comparable *MLL*bcr protection by Ku3 with and without the DNA-PK inhibitor (DNA-PKi) NU7441 (Supplementary Fig. 5a). When testing the effect of Ku3 on *MLL*bcr rearrangements in the absence and presence of DNA-PKi, we found significant ($p < 0.05$) and trendwise ($p < 0.10$), respectively, mitigation of *MLL*bcr rearrangements by Ku3. However, homologous DSB repair was augmented by DNA-PKi, likely due to a pathway shift, as seen after expression of I-SceI (Supplementary Fig. 5b, right panel). Such a pathway shift might introduce a confounding factor for the detection of the Ku3 effect. To circumvent a pathway shift during DSB repair, we performed measurements of NHEJ using a specific NHEJ reporter showing that Ku3, like Ku80-Ct, did not affect NHEJ frequencies (Supplementary Figs. 2c, 5c). Together with earlier screening data demonstrating that knockdown of Ku80 and Ku70, like EndoG, reduces *MLL*bcr rearrangements by 28-49%, while we saw a decrease by only 11% after knockdown of DNA-PKcs[6], our data strengthen the idea of an NHEJ- and at least partially DNA-PKcs-independent effect of Ku3 on *MLL*bcr rearrangements. These data show that the 28 aa peptide Ku3 derived from the C-terminal end of Ku80-Ct is sufficient to partially mitigate chemotherapy-related destabilizing effects on the *MLL*bcr region.

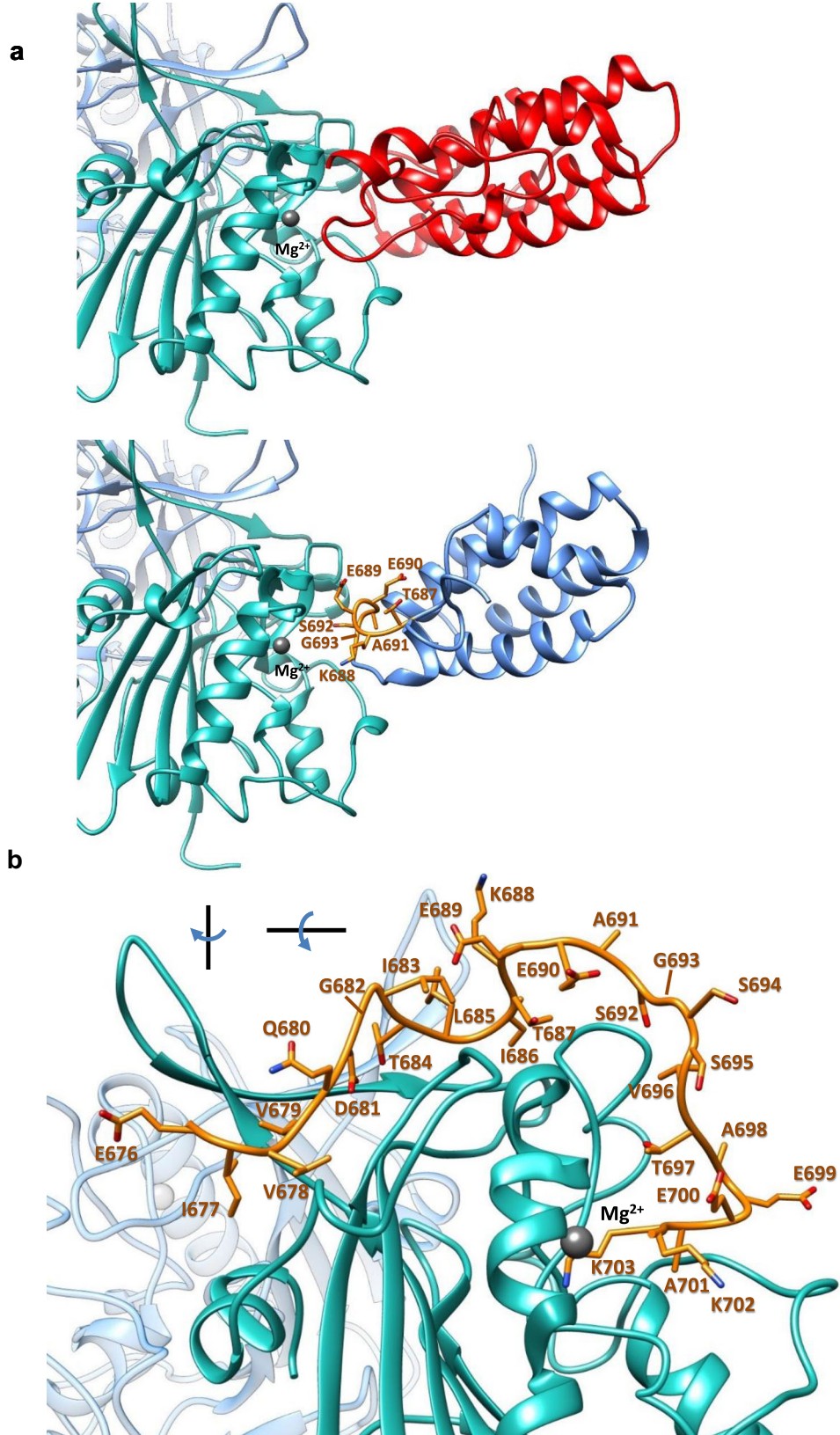

**Fig. 3 | Structure-based discovery of the Ku3 peptide. a** Top: Complex of EndoG (cyan)/EndoGI (red) from Drosophila melanogaster (PDB ID 3IMS). Gray sphere: Mg²⁺ ion. Bottom: Structure of the Ku80 fragment (light blue, residues 590–709, PDB ID: 1Q2Z), superposed on the EndoGI (transparent cartoon representation), bound to EndoG (cyan). Orange sticks: Fragment 687-693 (sequence TKEEASG) of Ku80, corresponding to the most conserved sequence motif among the extracted peptides. **b** Predicted docked complex of the Ku3 peptide (orange) bound to EndoG (cyan).

**Table 1 | List of selected Ku peptide candidates targeting EndoG and their properties**

| Peptide | Sequence | Molecular mass | Length (aa) | Residues in contact with EndoG | Fraction of peptide's length (% in contact) | Occupancy | Water solubility |
|---|---|---|---|---|---|---|---|
| Ku1 (aa 586–707) | ITKEEASGSSVTAEEAKKFLAP * | 2253 | 22 | 18 | 81.82 | 200 | Good/soluble |
| Ku2 (aa 599–624) | RVLVKQKKASFEEASNQLINHIEQFL | 3071 | 26 | 22 | 84.62 | 192 | Good/soluble |
| Ku3 (aa 676–703) | EIVVQDGITLLITKEEASGSSVTAEEAKK | 2933 | 28 | 18 | 64.29 | 179 | Good/soluble |
| Ku4 (aa 607–635) | ASFEEASNQLINHIEQFLDTNETPYFMKSI | 3518 | 30 | 18 | 60.00 | 145 | Poor/soluble |

**Peptide:** Ku80 peptides whose most populated cluster coincides with the lowest energy complexes are listed. The notation corresponds to the initial and final positions within the the aa sequence of human Ku80. EEA motifs are underlined.
**Residues in contact:** Number of residues in contact with the targeted binding region on EndoG (same binding region of the native Ku80).
Fraction of length (%): The percentage of the peptide length that is in contact with EndoG (contact distance <10 Å between the alpha carbons of the peptide and EndoG).
**Occupancy:** Number of structures in the cluster. 10 clusters out of 1000 structures were extracted. Average occupancy is 100, and extracted candidates are above the average.
**Water solubility** calculated using two tools: Innovagen/PROSO II-Innovagen (https://pepcalc.com/peptide-solubility-calculator.php) of N-terminal acetylated peptide -PROSO II ref. 39, https://febs.onlinelibrary.wiley.com/doi/10.1111/j.1742-4658.2012.08603.x.
* Note that the last aa of Ku1 (P) had to be replaced (G) to ensure efficient synthesis and purification of the peptide.

## Peptide Ku3 sequence-specifically interferes with Ku80 and EndoG complex formation in the nucleus

The identification of Ku80-Ct-derived peptide candidates was based on higher interaction scores with respect to the Ku80-Ct sequence, followed by further selection according to the docking simulations on the human EndoG model (Fig. 3). Consequently, we investigated whether the exogenous peptide Ku3 interferes with complex formation between endogenous Ku80 and EndoG. Focal accumulation of EndoG in the chromatin is rare even after doxorubicin treatment[30], which is in line with limited DNA strand breakage by EndoG in non-apoptotic cells[8]. Therefore, we visualized such Ku80-EndoG association by the sensitive in situ proximity ligation assay (PLA). As expected, only a few PLA foci indicating Ku80-EndoG complex formation were detected in HeLa reporter cells. Notably, PLA signals rose two- to three-fold after treatment with doxorubicin (Fig. 6a). When comparing cells nucleofected with 100 μg Ku3 with the mock controls, we observed a significant reduction of PLA foci numbers mediated by Ku3 (Fig. 6b). Such downregulation was not observed with 100 μg Ku1, the peptide sharing 18 aa with Ku3 but showing an anti-recombinogenic effect only at a higher level (Supplementary Fig. 4b). Direct comparison of cells nucleofected with Ku3 and with the scrambled version of Ku3 validated the inhibitory effect of Ku3, excluding a mere charge effect.

To gain further understanding of the interaction between EndoG and Ku3, we performed extensive molecular dynamics simulations, positioning the peptide far away from EndoG, for unbiased binding. Clustering analysis of the simulations showed that Ku3 binds at an EndoG region similar to the docking's predictions (Clusters 1 and 5, Supplementary Fig. 6a). In addition, Ku3 can bind regions close to the $Mg^{2+}$ cation in each EndoG monomer (Clusters 2 and 3, Supplementary Fig. 6b). Interestingly, these regions are also similar to the analogous binding interfaces in the Drosophila's EndoG-EndoGI complex[12] (Fig. 3a, top), which is also similar to the EndoG-Ku80 complex. Binding at these two regions could explain the inhibitory effects of Ku3 on both the activity of EndoG and binding to Ku80. The analysis of the simulations also showed that EndoG experiences major conformational fluctuations, especially upon Ku3 binding and as reflected by changes in the inter-chain angle ($\theta$) (Supplementary Fig. 6c). The analysis of the relationship between the distance ($d$) of EndoG and Ku3 center of masses (COM) and $\theta$, shows that upon Ku3 binding ($d <\approx$ 45 Å), EndoG undergoes more frequent and wider fluctuations of the inter-chain angle ($58° < \theta < 100°$), than when there is no binding ($d > 50$ Å, $65° < \theta < 75°$). This conformational flexibility could also influence the activity of EndoG in vivo.

Next, to further assess the impact of the peptide's sequence on binding to human EndoG, we generated 100000 Ku3-derivatives with a maximum of 10 mutations with respect to Ku3's sequence, using PPI-Detect. From this library, 18960 peptides showed an interaction score with EndoG equal or larger than that of Ku3. We then used these derivatives to generate a multiple sequence alignment (MSA) and identify conserved motifs determining peptide binding to EndoG. With this approach, we identified the motif EIVVQDGITLxxxxEx-EAxSGSSVxxExKK (The x denotes either gaps or non-conserved positions in the MSA), containing the submotif ExEAxSGSS, which we predicted to be important for peptide binding to EndoG with high specificity (Supplementary Fig. 7). This motif features two connected fragments: (1) a fragment containing two negatively charged residues followed by at least one small and hydrophobic residue [ExEAx], and (2) a fragment containing at least four small and polar residues [xSGSS]. With this result, we postulated that the fragment EEASGSS is the minimum active motif.

To test this hypothesis, we exchanged two conserved and differently charged amino acids derived from the 18 residues overlap between Ku3 and Ku1, namely E14 and K27 in Ku3 corresponding to E689 and K702 in Ku80 (Table 1 and Fig. 6c). Interestingly, both Ku3(E689A) (Mut1) and Ku3(K702A) (Mut2) not only abrogated

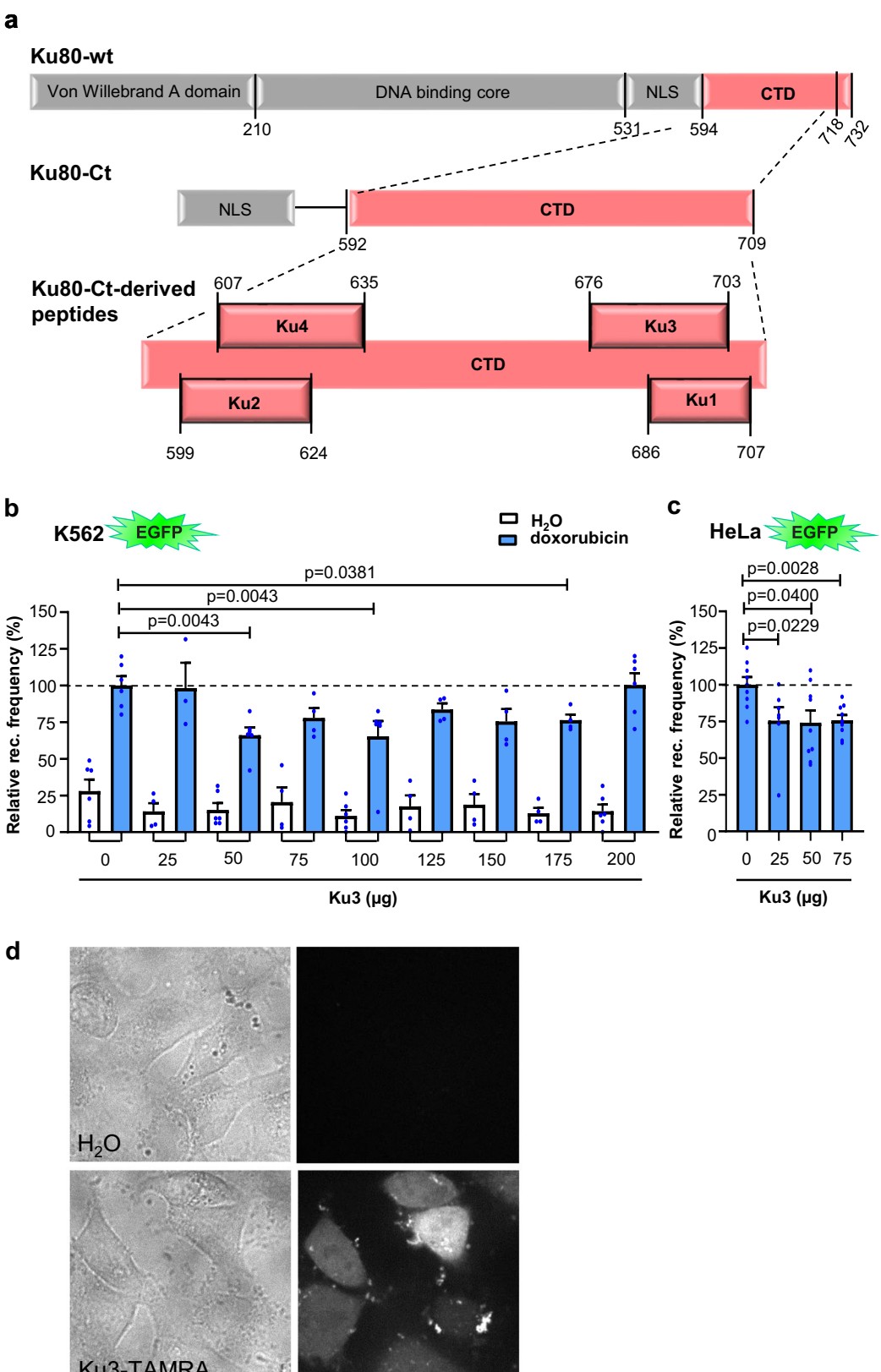

reduction of PLA foci numbers by Ku3 but in fact elevated these Ku80-EndoG specific signals. The residues $Ku3_{E14}$ ($Ku80_{E689}$) and $Ku3_{K27}$ ($Ku80_{K702}$) establish electrostatic interactions with $EndoG_{R100,\ R204,\ K214}$ and $EndoG_{E79,\ E112}$ respectively. Moreover, $Ku3_{D6}$ ($Ku80_{D681}$), which is missing in Ku1, establishes conserved electrostatic interactions with $Mg^{2+}$ and $EndoG_{R184}$ (Supplementary Fig. 8). Relative Ku3-specific

effects on Ku80·EndoG complex formation were similarly observed in untreated cells (Supplementary Fig. 9a–c). Further supporting the concept of replication-associated action of Ku80·EndoG complexes on the *MLL*bcr and related sequences under sublethal conditions[6,7], Ku80·EndoG PLA foci numbers rose 1.5-fold after aphidicolin treatment, inducing replication stress (Supplementary Fig. 9d). Our results thus

**Fig. 4 | Ku80-Ct-derived peptide Ku3 partially mitigates *MLL*bcr rearrangements. a** Scheme illustrating the positions of Ku peptides within Ku80-Ct. Nuclear localization signal (NLS); C-terminal domain (CTD, red). **b** Evaluation of peptide Ku3 in the *EGFP*-based reporter assay in K562. K562 reporter cells were nucleofected with increasing amounts of Ku3 peptide, cultivated for 24 h, treated for 4 h with 2 μM doxorubicin (blue) or $H_2O$ (white), released in drug-free medium for 72 h, recombination frequencies determined flow cytometrically and doxorubicin-treated cells without Ku3 set to 100% per experiment (average: $3 \times 10^{-5}$). Data are presented as mean + SEM; $H_2O$/doxorubicin, 0: $n = 6/6$; 25: $n = 4/3$; 50: $n = 6/6$; 75: $n = 4/4$; 100: $n = 6/6$; 125: $n = 4/4$; 150: $n = 4/4$; 175: $n = 4/4$; 200: $n = 6/6$ samples from 2 (25, 75, 125, 150, 175) or 3 (0, 50, 100, 200) independent experiments. Significances were calculated for mock- and doxorubicin-treated cells separately by Kruskal-Wallis H-test, followed by the two-tailed Mann-Whitney-U test and $p < 0.05$ indicated. For viabilities according to SSC/FSC gating, see Supplementary Fig. 4a. **c** Ku3 peptide evaluation in HeLa reporter cells. Increasing concentrations of the Ku3 peptide were nucleofected into cells, which were recultivated for 24 h, treated with 0.5 μM doxorubicin for 4 h and released in drug-free medium for 72 h. Recombination (rec.) frequencies were determined, presented, and statistics calculated as in (**b**); 0: $n = 9$; 25: $n = 7$; 50: $n = 9$; 75: $n = 9$ samples from 4 (25) or 5 (0, 50, 75) independent experiments. For viabilities according to SSC/FSC gating, see Supplementary Fig. 4f. **d** Cell entry of Ku3 peptide. Brightfield images of HeLa cells nucleofected with $H_2O$ or TAMRA-labeled Ku3 (left) and spinning disc confocal microscopy images at 532 nm laser excitation (right). Representative fields of view of > 20 cells. Scale bar is 20 μm. Source data are provided as a Source Data file.

indicate that Ku3 interferes with the formation of the complex of Ku80 with EndoG in the chromatin in a sequence-specific manner.

## Ku3 variants differentially affect EndoG binding to chromatin

Single-molecule tracking (SMT) provides direct insight into the behavior of proteins and enzymes in the environment of a living cell. In particular, SMT reports on kinetic parameters such as the mobility of molecules and their bound fractions and binding times to cellular structures. For example, the binding time is expected to differ between nonspecific, transient binding events and specific, productive interactions.

Using live-cell SMT, we aimed at quantifying the chromatin-bound fraction and chromatin residence time of EndoG compared to inactive control mutants and determine the changes in the presence of Ku80-Ct and Ku80 derived peptides. We therefore fused a HaloTag C-terminally to wild-type EndoG (EndoG-HT), the catalytically dead mutant EndoGΔcat (Supplementary Methods), which potentially exhibits longer residence times due to missing catalysis, and a monomeric mutant EndoG P199E (mEndoG)[11], which is expected to exhibit shorter residence times due to a lower affinity, and created stable HeLa cell lines by lentiviral transduction ("Methods"). We confirmed proper localization of the three EndoG-HT variants by visualization with HaloTag-SiR-ligand. They mainly localized to the mitochondria/cytoplasm and only sparsely in the nucleus (Supplementary Fig. 10a), in accord with the location of endogenously expressed EndoG[6]. We further quantified the expression of the EndoG-HT variants by western blotting, which revealed elevated expression levels compared to endogenous EndoG by factors of 4.5 for EndoG-HT, 3.4 for EndoGΔcat-HT, and 1.3 for mEndoG-HT (Supplementary Fig. 10b, c).

We first tested whether elevated expression levels of the Halo-tagged EndoG variants lead to binding saturation. Therefore, we imaged individual molecules labeled with SiR dye coupled to HaloTag ligand (HTL) at 19.3 Hz and detected and tracked single molecules using the software TrackIt[40]. We plotted the number of bound molecules tracked in at least two consecutive frames over the overall number of detected molecules (Supplementary Fig. 10d). A saturation of binding sites would be indicated if the curve approached a horizontal asymptote. However, we observed a close to linear increase, ruling out binding artifacts due to overexpression.

Next, we determined the overall bound fraction of EndoG-HT variants by imaging individual molecules labeled with HTL-PA-JF646 at 85 Hz (Fig. 7a)[41]. The histogram of frame-to-frame jump distances obtained from these tracks revealed an increase of short jumps for mEndoG compared to EndoG wt, while short jumps decreased for EndoGΔcat, indicating a lower bound fraction (Fig. 7b and Supplementary Fig. 11). Indeed, detailed analysis of the jump distance distributions using a model of three components of Brownian diffusion[40] confirmed a higher chromatin-bound fraction of mEndoG compared to EndoG wt, and a lower bound fraction for EndoGΔcat (Supplementary Fig. 12).

Third, we set out to quantify the residence time of the chromatin-bound fraction of EndoG variants. This analysis required monitoring chromatin-bound molecules over extended time periods. We therefore illuminated the cells with a time-lapse scheme including dark times of variable duration (Fig. 7c, d). The distribution of fluorescence survival times assembled from bound molecules indicated a faster decay and thus shorter residence time for mEndoG compared to EndoG wt, while EndoGΔcat showed on trend, a slower decay, indicating longer residence times (Fig. 7e). To obtain the residence times from the distribution of fluorescence survival times, we performed an inverse Laplace transformation, using the GRID method[42]. GRID yields a spectrum of dissociation rates that a molecular species experiences in the cell, which can be converted to residence times by inversion. The dissociation rate spectra of all EndoG variants consisted of several rate clusters (Fig. 7f and Supplementary Fig. 13). As expected from the distribution of fluorescence survival times, the rate cluster corresponding to the longest residence time was shifted to larger dissociation rates and thus shorter residence time for mEndoG compared to EndoG wt (Fig. 7f, g and Supplementary Table 2). Due to a lower affinity of the monomer to chromatin[11], shorter residence times are expected for monomeric compared to dimeric binding. In contrast, the residence time of EndoGΔcat did not markedly differ from the residence time of EndoG wt (Fig. 7g).

By combining the amplitude of the rate cluster corresponding to the longest binding times with the overall bound fraction, we obtained the long-bound fraction of EndoG variants (Fig. 7h). For mEndoG, the long-bound fraction was considerably higher than for EndoG wt, indicating a higher binding frequency of mEndoG. For EndoGΔcat, the long-bound fraction was comparable to the fraction of EndoG wt. Overall, our live-cell SMT experiments demonstrated that our assay is sensitive to differences in chromatin-bound fractions and chromatin residence times of EndoG, mEndoG and EndoGΔcat.

Next, we compared the chromatin-bound fraction and chromatin residence time of EndoG wt in the presence of Ku80-Ct. We therefore repeated our single-molecule measurements and analysis after transient transfection of HeLa cells stably expressing EndoG-HT with a plasmid expressing GFP-Ku80-Ct (EndoG + Ku80-Ct) and a GFP expression plasmid as control (EndoG transf. ctrl) (Supplementary Methods). Selecting for successfully transfected GFP-positive cells, we found similar residence times of EndoG in the presence and absence of Ku80 Ct (Fig. 7g and Supplementary Fig. 13), suggesting that, once bound, EndoG is not affected by the C-terminal domain of Ku80. However, the fraction of long EndoG binding events was reduced in the presence of Ku80-Ct compared to the transfection control (Fig. 7h and Supplementary Figs. 11, 12). This result corroborates our findings from immunofluorescence microscopic experiments, which revealed reduced chromatin binding of EndoG in the presence of Ku80-Ct.

Finally, we compared the chromatin-bound fraction and chromatin residence time of EndoG wt in the presence of the Ku80-Ct-derived peptides Ku3 and Mut1, and a Scrambled control. We again repeated our single-molecule measurements and analysis, now after nucleofection of TAMRA-labeled peptides (Supplementary Methods). Our experiments revealed comparable chromatin residence times for EndoG wt in the presence of Ku3 and Scrambled, while the residence

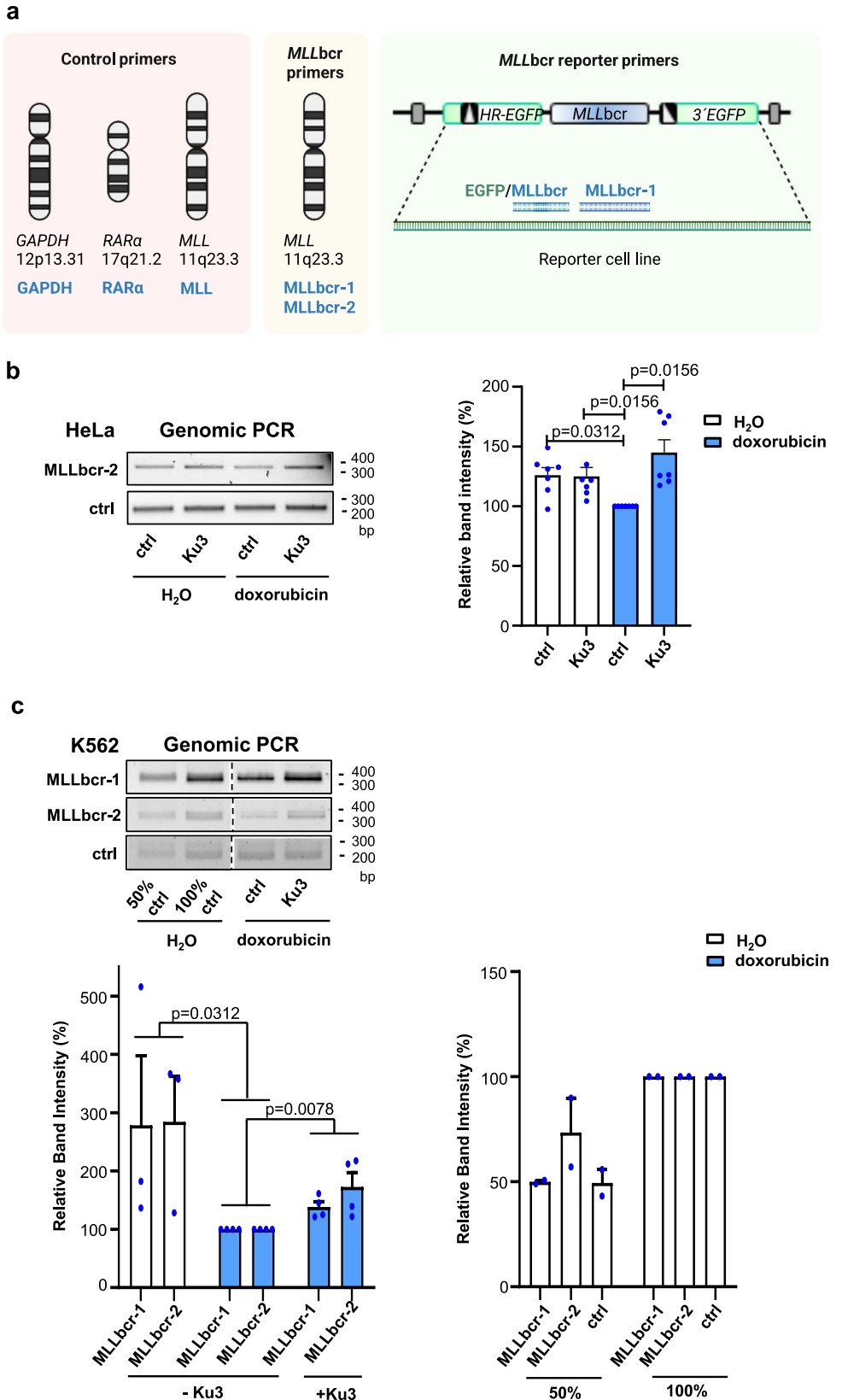

time dropped in the presence of Mut1 (Fig. 7g and Supplementary Fig. 13). Similarly, the long-bound fraction of EndoG wt was comparable in presence of Ku3 and Scrambled. In presence of Mut1, the long-bound fraction increased (Fig. 7h and Supplementary Figs. 11 and 12). Remarkably, while Ku3 did not alter the chromatin binding properties of EndoG wt, the behavior of EndoG wt in the presence of Mut1

mirrored the behavior of EndoG wt compared to mEndoG. Since our clustering analysis of the EndoG-Ku3 binding simulations (Supplementary Fig. 6) showed that Ku3 is theoretically able to contact several interfaces of EndoG and might increase the interchain angle of the EndoG dimer, it is tempting to speculate that Ku3 affects the activity of EndoG through modification of the dynamics of the EndoG dimer.

**Fig. 5 | Ku3 interferes with *MLL*bcr cleavage. a** Genomic PCR primer map. PCR primer sets (MLLbcr-1 and MLLbcr-2, blue, EGFP/MLLbcr, green/blue) were designed to amplify fragments sensitive to breakage at the therapy-related hotspot in the genomic *MLL*bcr locus (*MLL*bcr primers) and at the *MLL*bcr within the *EGFP*-based reporter construct chromosomally integrated in reporter cell lines (*MLL*bcr reporter primers). Control primer sets (ctrl: GAPDH, RARα, MLL, blue) targeting unaffected DNA loci *GAPDH*, *RARα*bcr and Intron20 in *MLL* were included to normalize and validate the results obtained from the *MLL*bcr (reporter) primers. Created in BioRender. Wiesmüller, L. (2026) https://BioRender.com/js049b4.
**b** Genomic PCR in Hela cells. Genomic PCR analysis was performed on HeLa reporter cells after nucleofection with 75 µg Ku3 and treatment with H$_2$O (white) or doxorubicin (0.5 µM, 4 h, blue), followed by agarose gel electrophoresis on Sea-Kem® LE Agarose (Lonza, Basel, Switzerland). The band intensities for MLLbcr-2 were corrected using MLL control PCR (ctrl) intensities and normalized to the doxorubicin treatment without peptide, set to 100%. Results are based on 7

independent experiments and presented as mean + SEM. Statistical analysis was conducted using the two-way Friedman test and two-tailed Wilcoxon matched-pairs signed rank test, and $p < 0.05$ indicated. **c** Genomic PCR in K562 cells. Genomic PCR analysis was performed with K562 reporter cells after treatment with H$_2$O or doxorubicin (2.0 µM, 4 h) and nucleofection with or without Ku3 (100 µg). PCR products were separated on UltraPure™ Agarose-1000 (Invitrogen/Thermo Fisher Scientific). The band intensities for *MLL*bcr using primer sets MLLbcr-1 and MLLbcr-2 were corrected each using control (ctrl) RARα PCR intensities and normalized to the samples with doxorubicin and without Ku3, set to 100%. Results are based on 3 (H$_2$O) to 4 (doxorubicin) independent experiments and results presented as means + SEM. Statistical analysis was conducted using a two-tailed Wilcoxon matched-pairs signed rank test. The panel on the right shows results from control amplifications engaging 50% and 100% template DNA from ctrl samples (H$_2$O) from 2 independent experiments. The stippled line indicates cropping. Source data are provided as a Source Data file.

## Discussion

The ancient nuclease EndoG needs to be well-controlled to protect cellular RNA and DNA from endonucleolytic attack and degradation, as indicated by the co-existence of naturally occurring endogenous inhibitory proteins in several lower organisms[43,44]. Such EndoG activity can cause rearrangements at the *MLL*bcr in human hematopoietic stem and progenitor cells, which are known to induce secondary leukemia in response to chemotherapeutic treatment[1,6,7]. Here, we show that the 3D structure of the C-terminal domain from the human DNA repair protein Ku80 closely resembles the EndoGI from Drosophila melanogaster. Our results from sequence alignments and molecular modeling reveal the binding interface of human EndoG between aa 110–184 with human Ku80-Ct. Peptides of 20–30 residues derived from Ku80 Ct were identified as candidates with larger predicted binding affinity than Ku80-Ct, clustering around residues 686–707 and to a lesser extent also around residues 599–624 within full-length Ku80. Our experiments validate these structure-based predictions, demonstrating that peptide Ku3, derived from the extreme C-terminal cluster in Ku80-Ct, reduces (i) Ku80-EndoG complex formation in the nuclear chromatin, (ii) breakage of the chromosomal *MLL*bcr region and (iii) rearrangements adjacent to the *MLL*bcr after treatment with the chemotherapeutic agent doxorubicin. Such a protective effect was found to be specific for the peptide sequence, it neither exhibited toxic effects nor inhibited the cytotoxic effect of the chemotherapeutic agent, presenting Ku3 as a highly interesting lead compound for further research aiming at preventing the development of secondary leukemia during cancer therapy.

This study also uncovers that Ku80-Ct resembles an EndoGI-like protein fragment, which is based on primary and 3D structure comparisons and is supported by Ku80-Ct´s antagonistic effect on treatment-induced *MLL*bcr destabilization, previously established to depend on EndoG[6,7]. The part of Ku80 corresponding to Ku80-Ct has been thought to serve as an extended, flexible arm that connects DNA-PKcs to DNA-bound Ku70-Ku80, thereby assembling the protein complex initiating NHEJ[45,46]. Our results provide evidence for the interference of Ku80-Ct with *MLL*bcr rearrangements independently of DSB repair, alluding to another role of Ku80 exerted via this domain. Thus, DSB repair triggered by targeted cleavage through I-SceI was not affected by Ku80-Ct, neither when engaging a reporter construct for NHEJ nor for homologous DSB repair without the therapy-responsive spacer *MLL*bcr. Moreover, Ku80-Ct, like Ku3, is devoid of the major binding sites for DNA, Ku70 and DNA-PKcs[31,32,46–48]. Finally, peptide Ku3 does not influence NHEJ but interferes with *MLL*bcr breakage independently of DNA-PK activity.

The ring-shaped Ku70-Ku80 complex serves as a scaffold for recruitment of repair factors, whereby the disordered Ku80 C-terminus has previously been considered suitable for interactions with other proteins[16]. The C-terminus of Ku80 was also reported to stimulate autophosphorylation of DNA-PKcs at specific sites, activating

processing of DNA ends via the DNA-PKcs complex partner Artemis[49]. Involvements of Artemis and of DNA-PK activity in *MLL*bcr rearrangements were excluded in our earlier study[7]. However, siRNA screening in this previous work revealed involvement of Ku80 in *MLL*bcr rearrangements together with EndoG[6]. Moreover, data from several studies suggest an overlapping set of proteins in the interactomes of EndoG and Ku80, including the base excision repair (BER) enzymes FEN1 and APE1[6,20,50,51]. APE1 promotes *MLL*bcr binding and cleavage by EndoG in response to replication stress, as demonstrated in our previous work[6]. Of note, the Ku heterodimer was reported to possess an intrinsic 5′ deoxyribosephosphate (5′dRP) and apurinic/apyrimidinic (AP) lyase activity, excising abasic sites such as generated by glycosylases from ends, analogous to that exhibited by DNA polymerase ß during BER[18]. EndoG and Ku80 are also both involved in class switch recombination (CSR) triggered by the DNA cytidine deaminase AID and downstream BER, preferentially at ssDNA targets of R-loops in G-rich DNA[52,53]. The G-rich *MLL*bcr has the propensity to form stable secondary structures and RNA-DNA hybrids, i.e., represents a potential off-target for such BER-coupled recombination processes[1,6]. Ku80, its binding partner Ku70 and EndoG are known to recognize G4 and other non-B-DNA secondary structures in the genome, such as cruciforms or hairpins[17,54–56]. Doxorubicin exacerbates the fragility of the *MLL*bcr through the generation of reactive oxygen species leading to clustered oxidative damage in such stable secondary DNA structures. As a DNA intercalator, it interferes with various DNA metabolic processes entailing transcription-replication conflicts and DSBs[1,6,26–28]. Our model implies Ku complex-mediated recruitment of EndoG to specific sites in the chromatin, where the Ku complex and EndoG synergistically recognize secondary structures such as predicted for the *MLL*bcr[1] and augment resolution of these replication barriers by EndoG.

In the light of these observations, our data suggest that Ku80 and EndoG might get trapped at the *MLL*bcr, possibly while cooperating during processing of abasic sites and/or when Ku modulates the order and extent of end resection[6,25,57–59]. We find that ectopic expression of Ku80-wt promotes specific binding of EndoG to the *MLL*bcr and local accumulation of γH2AX (Fig. 2), suggesting that Ku80 aids recruitment of nucleolytically active EndoG. Along this line, we previously showed that depletion of endogenous Ku80 reduces *MLL*bcr rearrangements[6]. Overexpression of exogenous Ku80-wt did not further impact on the integrity of the *MLL*bcr region (Figs. 1f, 2d), possibly reflecting a saturating effect of breaks in the *MLL*bcr or regarding the balance between Ku80 and EndoG[17]. What could be the biological meaning of such a sensitive and cooperative mechanism amplifying DNA damage selectively at fragile sites such as the *MLL*bcr? Distinct from EndoG´s canonical role in DNA fragmentation during the early stages of apoptosis, limited strand breakage by EndoG has been recognized as a source of mutagenesis and oncogenic transformation[8,10]. Similarly, the pro-apoptotic DNAse Caspase 3 was proposed to play a role in mutagenesis after

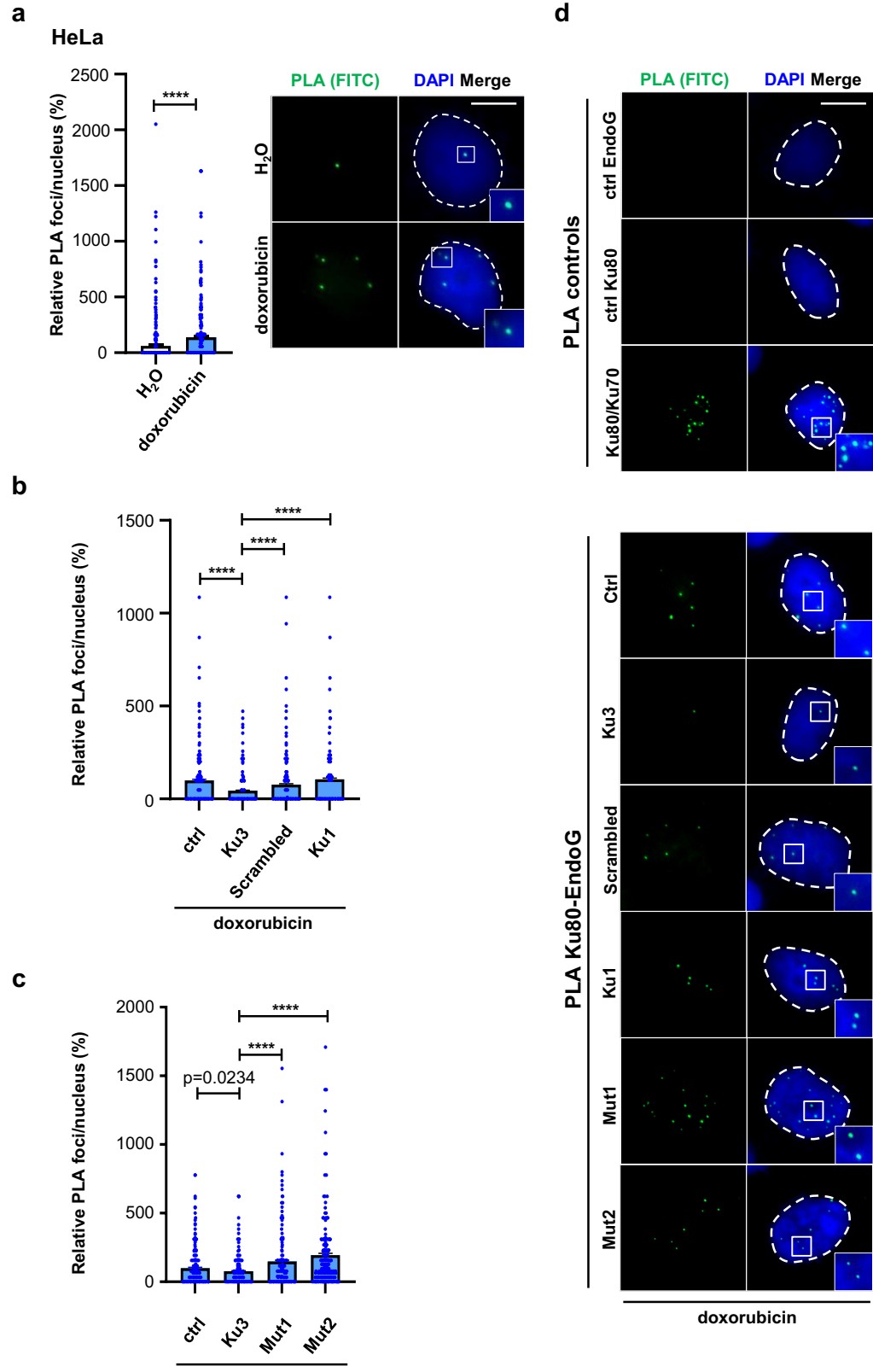

minor mitochondrial membrane permeabilization and sublethal Caspase 3 activation[2,8,10]. Considering recent work by Sørensen and colleagues on Caspase 3[60] nonlethal self-inflicted DNA strand breaks by such nucleases, including EndoG, may serve as early sensors to activate DNA damage response checkpoints and differentiation programs[61].

In this scenario, exogenous Ku80-Ct was expected to exert a dominant-negative effect on the crosstalk between endogenous Ku80 and EndoG, which was supported by the observed loss of EndoG foci in immunofluorescence microscopy (Fig. 2b), a reduced fraction of EndoG stably bound to the chromatin in SMT analysis (Fig. 7), as well as diminished breakage and rearrangements at the *MLL*bcr (Figs. 1f, 2d).

**Fig. 6 | Ku3 sequence-specifically interferes with the formation of complexes between Ku80 and EndoG.** Proximity ligation assay (PLA) was performed on fixed HeLa cells using antibodies against Ku80 and EndoG to detect Ku80-EndoG complex formation. **a** PLA of Ku80-EndoG complexes after doxorubicin treatment. HeLa cells were mock-transfected, cultured for 24 h, and treated with either doxorubicin (0.5 μM, 4 h, blue) or $H_2O$ (white). Foci numbers were normalized to the means of mock-transfected, doxorubicin-treated cells and set to 100% per experiment. Data are presented as mean + SEM from $n = 824$ nuclei for $H_2O$ and $n = 1200$ nuclei for doxorubicin from 9 independent experiments. Statistical significance was determined using the Kruskal-Wallis H-test followed by the two-tailed Mann-Whitney U test (****$p < 0.0001$). **b** Sequence-specific effect of Ku3 on Ku80-EndoG interactions after doxorubicin treatment. HeLa cells were nucleofected with Ku3, scrambled Ku3, or Ku1 peptide, cultured for 24 h, and treated with doxorubicin (0.5 μM, 4 h). PLA analysis was performed, and foci numbers were normalized to the means of control cells (no peptide transfected) and set to 100% per experiment (average: 2 foci/nucleus). Data are presented as mean + SEM from $n = 596$ nuclei for ctrl, $n = 761$ nuclei for Ku3, $n = 659$ nuclei for Scrambled and $n = 445$ nuclei for Ku1 from 4 independent experiments. Statistical significance was determined using the Kruskal-Wallis H-test followed by the two-tailed Mann-Whitney U test (****$p < 0.0001$). **c** Ku3, but not its mutated versions, affects Ku80-EndoG interactions after doxorubicin treatment. HeLa cells were nucleofected with Ku3 or two mutated versions of Ku3 (Mut1 and Mut2), cultured for 24 h, and treated with doxorubicin (0.5 μM, 4 h). PLA analysis was performed, and foci numbers were normalized to the means of control cells (ctrl, no peptide transfected) and set to 100% per experiment. Data are presented as mean + SEM from $n = 401$ nuclei for ctrl, $n = 536$ nuclei for Ku3, $n = 404$ nuclei for Mut1 and $n = 401$ nuclei for Mut2 from 3 independent experiments. Statistical significance was determined using the Kruskal-Wallis H-test followed by the two-tailed Mann-Whitney U test (****$p < 0.0001$). **d** PLA image gallery. As PLA controls, we included primary antibodies against Ku80 or EndoG (negative controls) or antibodies against Ku80 and Ku70 (positive control). Representative PLA images were derived from 4 and 3 independent experiments in (**b**) and (**c**), respectively. Margins of DAPI-stained nuclei are indicated by white stippled lines, and the scale bar represents 10 μm. Insets display highlighted regions at two-fold magnification. Source data are provided as a Source Data file.

Importantly, Ku80-Ct did not alter doxorubicin-induced toxicities, as deduced from FACS, DNA content, MTT and pan-nuclear γH2AX analyses (Supplementary Figs. 1–3). Taken together, Ku80-Ct possesses the predicted features of a human EndoGI antagonizing the *MLL*bcr destabilizing effect of EndoG during chemotherapeutic treatment, but without affecting cell killing.

In mitochondria, EndoG levels are high (Supplementary Fig. 10a) and an alternate, C-terminally truncated form of Ku80 is found[62]. However, a Ku80-dependent control mechanism of EndoG may not be necessary in mitochondria, because EndoG resides in the inter-membrane space, i.e., is spatially separated from the mitochondrial genome, at least in unstressed cells[63]. Due to the increased oxidative environment, doxorubicin-induced DNA damage is also elevated in mitochondria[26], and our recent works revealed a role of EndoG in nucleolytic removal of irreparably oxidatively damaged copies[64,65]. However, multiple remaining copies of the mitochondrial genome and compensatory replication of these intact copies ensure mitochondrial genome homeostasis[64,65]. Altogether, while the nucleolytic activity of human EndoG in the nucleus must be tightly controlled to maintain genomic integrity, proposedly through interactions with the EndoGI-like Ku80 C-terminal domain, such a control mechanism might be obsolete in mitochondria.

The peptide Ku3, identified in silico, functionally mimicked EndoGI features of Ku80-Ct (Figs. 3–5 and Supplementary Fig. 6). The predicted EndoG/Ku3 binding interface provides a molecular rationale for the effect of Ku3 as a competitive inhibitor of EndoG-Ku80 interactions. This competitive mode-of-action is experimentally supported by the inhibitory effect of Ku3 on Ku80-EndoG complex formation (Fig. 6). Moreover, loss of such inhibition was detectable for Ku3-derivatives mutated at positions E14 (E689 in Ku80) or K27 (K702 in Ku80) of the conserved motif identified in-silico. The analysis of extensive molecular dynamics simulations delivers a better understanding of the binding and dynamics of Ku3 with EndoG in solution. These simulations confirmed the binding mode predicted by docking calculations and revealed other binding regions at EndoG that further supported the competitive mode-of-action. Experimental support for Ku3 interactions with EndoG that cannot fully be explained by competition with Ku80 might also be extracted from our experimental observations with mutated peptides. Thus, interactions with mutated Ku3 derivatives not only lost the inhibitory effect on the Ku80-EndoG interaction but also even promoted complex formation (Fig. 6 and Supplementary Fig. 9). Importantly, our simulations showed that Ku3 also interacts with the EndoG dimer through coordination of the $Mg^{2+}$ cation and interactions with its adjacent residues. Additionally, binding of Ku3 modifies the dynamics of the EndoG dimer, with an increase of the inter-chain angle, which may impact EndoG activity. Our work

hints at the possibility that EndoG is susceptible to an allosteric regulation that may affect DNA binding directly or indirectly through conformational changes. SMT experiments indicated reduced affinity of EndoG chromatin interactions in the presence of mutated versus wild-type Ku3. We hypothesize that peptide Mut1 has lost interactions competing with EndoG-Ku80 complex formation but retains the capacity to promote structural changes of EndoG. This information can be exploited for the design of specific EndoG inhibitors in the future. Furthermore, Ku80-Ct interacts with EndoG through multiple regions along its sequence clustering around residues 686–707 and to a lesser extent also around residues 599–624 (Fig. 4a). Future investigations might therefore explore the possibility of combining peptides through a dual spacer to optimize binding affinities and specificities.

In summary, our work describes a peptide inhibitor of EndoG that selectively blocks its pro-tumorigenic functions but not its pro-apoptotic ones, unlike the previously published chemical inhibitors non-selectively inhibiting EndoG[66]. Nonspecific effects on EndoG´s nuclease activity may prevent the cytotoxic effect of EndoG during chemotherapy, which undesirably mediates resistance of tumor cells. Targeted interference, which selectively blocks only certain molecular interactions, is difficult to achieve with small molecules[67]. Here, we employed computational modeling to identify peptides that are highly specific for the molecular target EndoG and therefore protective against *MLL*bcr rearrangements during replication stress. Preventive combination therapy with such peptides or derivatives may allow the reduction of a severe side effect of chemo- and radiotherapies, namely the development of secondary tumors, especially therapy-related ALL, therapy-related myelodysplastic syndrome or therapy-related AML.

## Methods
### Molecular modeling
**Homology modeling of human EndoG.** The Swiss-Model server[68] was used to generate a homology model of the human EndoG (Uniprot ID: Q14249) based on 6 templates: 4QN0_A, 3ISM_A, 3ISM_B, 3S5B_A, 3S5B_B, and 5GKC_A. From these templates, nine homology models were built, and the best one, based on the QMEAN quality assessment[69] was selected.

**In silico screening of Ku80-based peptide candidates.** We used the sequence of the C-terminal domain of Ku80 (Ku80-Ct, PDB ID: 1Q2Z[36]) to generate a library of 5050 Ku80-Ct-derived peptides with more than 20 residues. We screened these peptides for their interaction with human EndoG with PPI-Detect[37]. PPI-Detect is based on a support-vector machine model that predicts the interaction likelihood between sequences. We selected the peptides with an interaction score higher

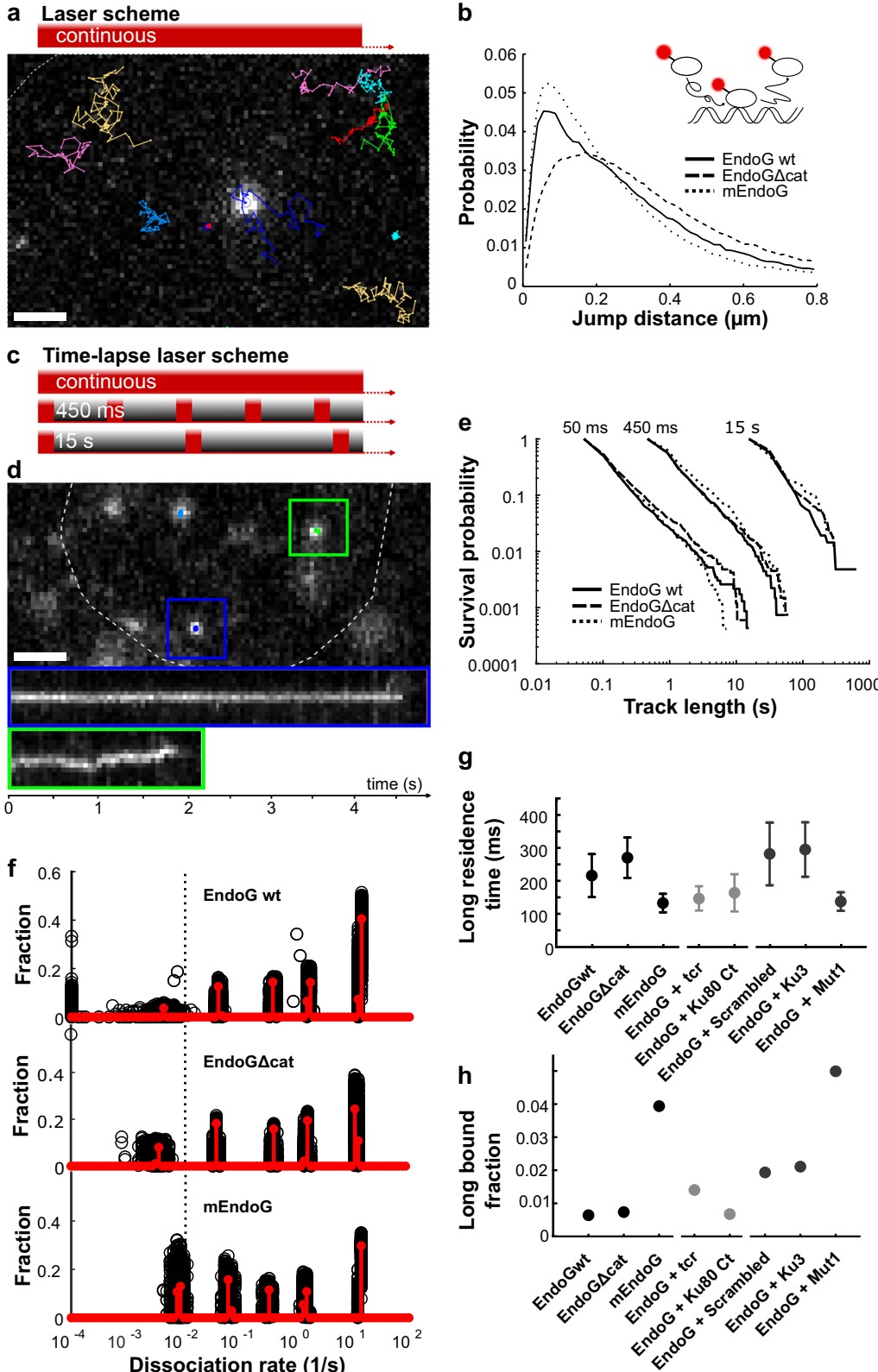

**a** Laser scheme

**b**

**c** Time-lapse laser scheme

**d**

**e**

**f**

**g**

**h**

than the full-length Ku80, resulting in 356 peptides with sizes between 20–60 residues.

## Molecular docking

From the set of 356 peptides identified with higher interaction scores than the full Ku80, we extracted all the candidates with 30 or fewer residues. Such selection reduced the set of candidates to 137 peptides.

All these peptides were docked with the EndoG model using the program CABS-Dock[38]. We performed 10 replicas of the algorithm, extracting the 100 top-scored binding poses from each replica. The obtained 1000 structures were clustered into 10 clusters using the k-medoids algorithm. For each peptide, the lowest energy and the most populated clusters were identified and extracted as the most suitable binding poses. To further filter the ranked docked candidates,

**Fig. 7 | SMT: peptides differentially affect EndoG binding to chromatin. a** Upper panel: scheme of illumination pattern for EndoG bound fraction analysis. Lower panel: example tracks of EndoG wt recorded at 85 Hz, overlaid with an example frame of the corresponding movie. Scale bar is 2 μm. **b** Jump distance distributions of EndoG variants. Inset: Sketch of EndoG-HT diffusion and DNA binding. **c** Scheme of illumination pattern in time-lapse measurements with indicated frame cycle times. **d** Example tracks of EndoG wt recorded with 51.7 ms frame cycle time, overlaid with an example frame of the corresponding movie, and kymographs of indicated molecules. Scale bar is 2 μm. **e** Survival time distributions of EndoG variants at time-lapse conditions shown on top. **f** State spectra of dissociation rates of EndoG variants obtained with GRID using all data (red). As an error estimation, GRID was run 500 times, with each run using 80 % of the data (black circles). The

dotted line indicates the dissociation rate of 0.01 s$^{-1}$. **g** Long residence times (inverse of the weighted average of dissociation rates below 0.01 s$^{-1}$ in (f)) of EndoG variants and EndoG wt in the presence of a transfection control plasmid (tcr), Ku80 C-terminus, and Ku3-derived peptides. Error bars denote SD of the resampled data. **h** Long bound fractions of EndoG variants, and EndoG wt in the presence of tcr, Ku80 C-terminus and Ku3 derived peptides calculated from amplitudes in (**f**) and Supplementary Fig. 13b and bound fractions in Supplementary Fig. 12b. Data represented as value ± SD from Gaussian error propagation. The error bars are smaller than the data markers and are therefore not visible. Source data are provided as a Source Data file. Measurement statistics are given in Supplementary Table 3.

we computed the number of residues of the peptides interacting with EndoG and re-ranked according to the fraction of their length that it is in contact with EndoG in the most stable complex and/or the most populated one. A threshold of 50% of the peptide length was used to extract the peptides with the largest binding interface with EndoG. This analysis resulted in 13 peptides with a large contact area with the targeted binding site on EndoG (Table 1) that were selected for experimental testing.

**Identification of Ku3-derived minimal binding motif.** Starting from the Ku3 sequence, 100000 derivatives, each one with a maximum of 10 modifications, were screened using PPI-Detect[37]. From them, 18960 peptides showed an interaction score with EndoG equal to or larger than that of Ku3. These peptides were selected to build a multiple sequence alignment (MSA) to identify conserved motifs that can be responsible for the observed binding. The programs MAFFT (v7)[70], MaxAlign[71], and EMBOSScons[72] were used to align, refine and extract conserved motifs, respectively. Consensus positions were accepted if they have positive matches (plurality) of at least half of the entries in the MSA. A WebLogo plot[73,74] was used to visualize the frequency of appearance of each residue per peptide position.

**Molecular dynamics simulations.** Based on our docking model of EndoG/Ku3, three configurations were prepared where the peptide was separated from the EndoG dimer at distances between their centers of masses (COM) of 61.75 Å, 64.35 Å, and 50.21 Å. We thus ensured that the side chains were sufficiently separated to avoid any intermolecular interaction in the starting structures. CHARMM-GUI[75] was employed to prepare the systems to run with GROMACS[76] using the CHARMM36m force field[77]. EndoG and Ku3 were inserted in a cubic simulation box with 15 Å of padding. The box was solvated with the TIP3P water model[78] and NaCl at 150 mM. After minimization and equilibration, three replicas of 1 μs each per starting configuration (9 replicas in total, each 1 μs) of production runs were performed, with a time step of 2 fs. Neighbor searching was performed every 20 steps. The PME algorithm[79] was used for treating electrostatic interactions with a cutoff of 1.2 nm. A single cutoff of 1.225 nm was used for Van der Waals interactions. Temperature coupling was done with the Nose-Hoover[80] algorithm. Pressure coupling was done with the Parrinello-Rahman[81] algorithm. The analyses were performed using the last 700 ns of each replica (totaling 6.3 μs) using VMD[82], CONAN[83], and GROMACS's tools. Conformational clustering of the Ku3 peptide was performed with the VMD plug-in (https://github.com/luisico/clustering), employing an RMSD cutoff of 20 Å for the backbone atoms of the peptides. The distance between the molecules ($d$) was defined as the distance between their centers of masses (COM). The EndoG interchain angle ($\theta$) was defined by the Cα atoms of EndoG-A$_{V95}$, EndoG-A$_{K214}$, and EndoG-B$_{V95}$ residues. The frequency of the $d/\theta$ pairs and the density plot were calculated with 2D binning of 151 × 151 points with OriginPro (v2022b) with the default settings. For the contact analysis with CONAN, a distance cutoff of 5 Å between the residue's heavy atoms and a lifetime of ≥ 50 % was employed.

## Cell culture and treatments
K562 cells (Cellosaurus RRID:CVCL_0004, provided by Heinrich-Pette-Institute, Hamburg, Germany) and the reporter cell line derivative K562MLL, stably transfected with the DNA recombination substrate pHR-EGFP/3'EGFP-MLLbcr.fwd[29], were cultured in RPMI 1640 medium (Gibco/Thermo Fisher Scientific, Waltham, Massachusetts, USA) supplemented with 10% fetal bovine serum (Pan-Biotech GmbH, Aidenbach, Germany) and 1% L-glutamine (Gibco/Thermo Fisher Scientific). HeLa cells (Cellosaurus RRID:CVCL_0030, provided by Heinrich-Pette-Institute, Hamburg, Germany) and the reporter cell line derivative HeLaMLL[7] were cultured in DMEM (Gibco/Thermo Fisher Scientific) supplemented with 10% fetal bovine serum (Biochrom, Berlin, Germany) and 1% L-glutamine (Gibco/Thermo Fisher Scientific). Culturing was performed in a humidified incubator at 37 °C with 5% CO$_2$. All cell lines were tested for mycoplasma contamination by PCR and confirmed to be negative. All human cell lines were authenticated by Microsynth AG, Balgach, Switzerland, using STR profiling within the last three years.

Twenty-four hours after seeding in a 6-well suspension cell culture plate (SARSTEDT AG & Co. KG, Nümbrecht, Germany), 24 h after plasmid electroporation and 24 h after peptide nucleofection, respectively, K562 (reporter) cells were treated for 4 h with either H$_2$O or 2.0 μM doxorubicin (University Hospital Ulm) in H$_2$O or with either DMSO or 10 μM etoposide (Sigma-Aldrich/Merck, St. Louis, Missouri, USA). Twenty-four hours after seeding in a 6-well adherent cell culture plate (SARSTEDT AG & Co. KG), 24 h after plasmid transfection and 24 h after peptide nucleofection, respectively, HeLa (reporter) cells were treated with H$_2$O or 0.5 μM doxorubicin in H$_2$O for 4 h, except for viability analysis 24 h after doxorubicin treatment.

For FACS-based recombination measurements, reporter cells were washed in Phosphate Buffered Saline (PBS) and re-cultivated in fresh medium for 72 h before harvest. For immunofluorescence microscopy cells were fixed, for ChIP or genomic DNA isolation cells were harvested immediately after treatment. For co-treatment with doxorubicin and DNA-PK inhibitor, HeLa reporter cells were treated with either H$_2$O or 1 μM NU7441 (STEMCELL Technologies, Vancouver, Canada) for 1 h before being exposed to H$_2$O or 0.5 μM doxorubicin in H$_2$O for 4 h.

## Plasmids, electroporation and transfection
For analysis of Ku80-wt, Ku80-Ct or Ku80(E720A, E721A) the following expression plasmids with a pCMV6-AN-DDK backbone were generated (OriGene, Rockville, Maryland, USA): pCMV6-AN-DDK-Ku80 expressing Ku80-wt (SC112985 ORF, OriGene) with N-terminal Flag-tag and pCMV6-AN-DDK-Ku80 C-terminus expressing Ku80-Ct through fusion of the Ku80-NLS motif P562-K568 (encoded by 5′-CCTACAGCTAAAAAATTAAAG-3′).

For homologous DSB repair and NHEJ measurements, we engaged reporter plasmids p5xtrshSV40HygbPuroCMV-HR and EJ5SceGFP, respectively, together with pCMV-I-SceI for targeted cleavage by the meganuclease I-SceI[34,84]. Plasmid p5bPuroCMV-wtEGFP, generated for

EGFP expression in the reporter plasmid backbone, served to determine transfection efficiencies.

To introduce protein expression plasmids into K562 (reporter) cells, these suspension culture cells were split, cultured for 24 h, washed in RPMI 1640 medium, no glutamine, no phenol red (Gibco/Thermo Fisher Scientific) and resuspended in 400 μl of the same medium. Then, a total amount of 50 μg plasmid for expression of one of the Ku80 variants was added, and the mixture transferred into a Gene Pulser®/Micropulser™ electroporation cuvette (Bio-Rad Laboratories, Hercules, California, USA), followed by electroporation at 200 V and 1050 μF with a Gene Pulser Xcell™ electroporation system (Bio-Rad Laboratories) and recultivation in fresh medium. For homologous DSB repair or NHEJ measurements, K562 cells were electroporated with 40 μg of the Ku80 variant expression plasmid, 10 μg of p5xtrshSV40HygbPuroCMV-H or EJ5SceGFP reporter plasmid, 10 μg of pCMV-I-SceI and 10 μg of filler plasmid pBS (pBlueScriptII KS (+/-) from Stratagene, Heidelberg, Germany), the latter of which was replaced by p5bPuroCMV-wtEGFP in transfection controls. For homologous DSB repair measurements, K562 cells were recultivated for 48 h, for protein isolation and Western blotting for 24 h.

HeLa (reporter) cells were transfected using FuGENE®HD (Promega, Fitchburg, Madison, USA) 24 h after seeding, whereby a mixture of 1.940 ml OptiMEM (Thermo Fisher Scientific), 20 μg plasmid DNA and 40 μl FuGENE® HD solution were dripped on the cells, followed by 6 h incubation, a medium change and recultivation for 16 h.

## Western blot

Cell lysates were prepared from scraped HeLa or centrifuged K562 cells by incubation in lysis buffer (50 mM Tris-HCl, Tris-Base, pH 7.4, 150 mM NaCl, 2 mM EGTA, 2 mM EDTA, 25 mM Sodium fluoride, 25 mM ß-Glycerol phosphate, 0.1 mM Sodium vanadate, 0.2% Triton X-100 0.3% Nonidet P40, protease inhibitor cocktail tablet, PIC, Sigma-Aldrich/Merck St. Louis, Missouri, USA) for 30 min on ice. After clearing the suspension by centrifugation (15 min, 1380 x $g$, 4 °C), the supernatant was subjected to determination of protein concentrations by use of the BCA Protein Assay Kit (Thermo Fisher Scientific) and to denaturing SDS-polyacrylamide gel electrophoresis, Western transfer and immunodetection of transferred proteins, as described before[7]. Primary antibodies for immunodetection were: anti-Ku80 (rabbit polyclonal, H-300, sc-9034, Santa Cruz Biotechnology Inc., 1/500), anti-Flag M2 (mouse monoclonal, F1804, Sigma-Aldrich/Merck, St. Louis, Missouri, USA, 1/1000), anti-Flag M2-Peroxidase (HRP, mouse monoclonal, A8592, Sigma-Aldrich/Merck, St. Louis, Missouri, USA, 1/1000), anti-EndoG (Mouse monoclonal, B-2, sc-365359, Santa Cruz Biotechnology Inc., 1/500), anti-γH2AX Ser139 (Mouse, monoclonal, Clone JBW 301, 05-636, Merck Millipore, Burlington, Massachusetts, USA, 1/1000), anti-DNA-PKcs phospho S2056 (Rabbit, polyclonal, ab18192, Abcam, 1/1000), anti-DNA-PKcs (mouse monoclonal, ab1832-500, Abcam, 1/200), anti-Vinculin (mouse monoclonal, V9131, Sigma-Aldrich, or mouse, monoclonal, sc73614, Santa Cruz Biotechnology Inc., 1/500) and anti-Tubulin (mouse monoclonal, ab7291-100, Abcam, 1/5000). Secondary antibodies were: horseradish peroxidase (HRP)-conjugated goat-anti-mouse (610-1319, Rockland, Pennsylvania, USA, 1/10000) and goat-anti-rabbit (611-1322, Rockland, Pennsylvania, USA, 1/10000). Western blot signals were visualized by Super Signal West Dura Extended Duration Substrate (Thermo Fisher Scientific) and a ChemiDocTM MP Imaging System (Bio-Rad Laboratories, Hercules, California, USA). Quantification of band intensities was performed with Image Lab 5.2.1 (Bio-Rad Laboratories) and corrected with the values of the corresponding anti-Tubulin loading controls.

## FACS-based recombination, cell cycle and viability analysis

Determination of treatment-induced intrachromosomal recombination frequencies as well as extrachromosomal homologous DSB repair and NHEJ measurements upon expression of meganuclease I-SceI were performed as described previously[6,30]. Briefly, fractions of EGFP-positive cells among 0.5-1.0 million viable cells were measured flow cytometrically by use of a FACSCalibur™ and BD CellQuest Pro 5.2.1 (BD Biosciences, San Jose, California, USA) or CytoFLEX B3-R1-V0 flow cytometer with APD detectors and GFP-oD1 bandpass filter and CytExpert, Version 2.5.0.77 (Beckmann Colter, Brea, CA, USA), as indicated. Hereby, viable cells were selected by the SSC/FSC gate and green-fluorescent cells therein detected in the diagonal auto-fluorescence-FL2/EGFP-FL1 dot plot gate (Supplementary Figs. 1, 2). To correct for interexperimental variations, mean recombination frequencies in appropriate controls were set to 100% in each experiment, and relative percentages calculated for each single value.

DNA content analysis was performed as before, applying the DNA intercalating agent propidium iodide (PI)[6,30]. Briefly, cells were washed with PBS, ice-cold fixative (80% ethanol/100% acetone 1:1) added slowly and cells incubated for ≥ 15 min at − 20 °C, washed first in PBS/fixative (1:1) and then in PBS. In the dark, cells were resuspended in 200 μl PI solution (1/20 PI stock solution from Sigma-Aldrich/Merck; 1:400 50 μg/ml RNaseA from OriGene in PBS), incubated at 37 °C for 30 min and stored at RT until FACS analysis (FACSCalibur™, BD Biosciences) to quantify the distribution of viable cells in different cell cycle phases and the percentage of apoptotic cells with sub-G1 DNA content.

Viabilities of HeLa cells treated with doxorubicin were assessed by MTT (3-(4,5-dimethylthiazol-2-yl)-2,5-diphenyl tetrazolium bromide; Sigma-Aldrich/Merck, St. Louis, Missouri, USA) assay as before[29]. Optical densities were measured with a Tecan Sunrise Photometer (Tecan, Crailsheim, Germany) at 570 nm.

## Genomic polymerase chain reaction (PCR)

Genomic DNA was isolated using the QIAamp DNA Mini Kit (Qiagen, Hilden, North Rhine-Westphalia, Germany) according to the manufacturer's instructions. PCR was performed with 20 ng DNA in a total reaction volume of 30 μl including 1.5 mM MgCl$_2$ for MLLbcr-2 and 1.0 mM MgCl$_2$ for MLL (Intron20 in *MLL*) primer sets when using my-Budget PCR kit (Bio-Budget, Krefeld, Germany) or with 20 ng DNA in a total reaction volume of 50 μl when using Platinum™ II Hot-Start PCR Master Mixes (2X) (Invitrogen/Thermo Fisher Scientific, Waltham, Massachusetts, USA) in a FlexCycler (Analytik Jena, Jena, Germany). PCR products were separated by agarose gel electrophoresis using SeaKem® LE Agarose (Lonza, Basel, Switzerland, 2.5%) or UltraPure™ Agarose-1000 (Invitrogen/Thermo Fisher Scientific, 3%) and visualized by SYBR™ Safe DNA Gel Stain (Invitrogen/Thermo Fisher Scientific). To ensure the reliability of the PCR conditions, two technical controls were included in each experiment. First, to verify that the selected cycle number was within the linear range, a control sample containing 50% of the DNA of the untreated sample was included. Second, a DNA-free control was included to check for potential contamination, i.e., a control sample containing only the PCR master mix without DNA. The primers (Biomers.net, Ulm, Germany) amplifying *MLL*bcr-specific and control fragments had the following sequences (Fig. 5a): MLLbcr-1 (351 bp fragment at the intron11-exon12 boundary surrounding the major therapy-related hotspot *MLL*bcr), FWD: 5´-GGCTCACAACA-GACTTGGCAAT-3´, REV: 5´-TATTTCCCCCACCCCACTCC-3´; MLLbcr-2 (345 bp fragment, 0.8 kb downstream of the therapy-related hotspot *MLL*bcr within the *MLL* breakpoint region), FWD: 5´-TTGGGTGTAAT-CAGTTGCCTATT-3´, REV: 5´-GGGTGATAGCTGTTTCGGCA-3´; EGFP/MLLbcr (261 bp fragment of the *MLL*bcr-reporter across the boundary of the *HR-EGFP* cassette and the *MLL*bcr spacer), FWD: 5´-ACCAC-TACCAGCAGAACACC-3´, REV: 5´-ATACGAAACAGTTGTAAGTATCG TC-3´; MLL (236 bp fragment of the genomic *MLL* intron 20 used as a control in genomic PCR experiments): FWD: 5´-GCAACA-CAGGGCCCTAGTTAAT-3´, REV: 5´-AGGCAAATCAGTCACCTTTTAAT CA-3´; RARα (205 bp fragment of the *RARα*bcr used as a control in PCR), FWD: 5´-TCCCTCCTCTTCAAGCGTTA-3´, REV: 5´-CCCATCACTC

CTCTGGACTC-3´; GAPDH (294 bp fragment of the *GAPDH* gene promotor, used as a control in PCR), FWD: 5´-CCCAACTTTCCCGCCTCTC-3´, REV: 5´-CAGCCGCCTGGTTCAACTG -3´; The band intensities of *MLL*bcr-specific amplifications (MLLbcr-1, MLLbcr-2 and/or EGFP/MLLbcr) were corrected using band intensities of control PCR reactions (ctrl: GAPDH, RARα and/or MLL) from the same DNA sample and then normalized to bands from appropriate internal controls run in each experiment and set to 100%, such as obtained from samples treated with doxorubicin but without peptide. Images of agarose gels were recorded by ChemiDoc™ MP Imaging System (Bio-Rad Laboratories, CA, USA) and band intensities quantified in the linear range using Image Lab 5.2.1 to 6.0.1 (Bio-Rad Laboratories).

### Chromatin immunoprecipitation (ChIP)

Chromatin immunoprecipitation was performed as detailed previously[6,30]. All steps and buffers used until the washing step of the immunoprecipitate were done on ice or at 4 °C. First, HeLa cells were fixed with 1% formaldehyde in medium for 10 min followed by 5 min quenching with 0.12 M glycine in medium on a shaker. Thereafter, cells were washed two times with ice-cold PBS, scratched off with cell scraping solution (PBS with 5 µl/ml phenylmethylsulfonyl fluoride, PMSF, Sigma-Aldrich/Merck), centrifuged and resuspended in 1 ml cell scraping solution and centrifuged. For DNA and protein extraction, the cells were resuspended in 1 ml ice cold cell lysis buffer (5.0 mM PIPES, pH 8.0; 167 mM NaCl; 1.2 mM EDTA; 0.01% SDS; 1.1% Triton X-100) with 5 µl PIC and 5 µl PMSF. After 10 min of incubation, the cells were dounced, centrifuged and broken cells dissolved in 1 ml ice cold nuclei lysis buffer (50 mM Tris-HCl, pH 8.0; 10 mM EDTA; 1.0% SDS) with 5 µl PIC and 5 µl PMSF and transferred into a polystyrol tube. After 10 min of incubation, the cells were sonified 5×15 s with 1 min breaks. Thereafter, 50 µl were taken for the input sample, and the remaining suspension frozen in liquid nitrogen and stored at − 80 °C. The input sample was mixed with 150 µl diethyl pyrocarbonate (DEPC) water and 10 µl 5 M NaCl and incubated overnight at 65 °C, then incubated with 0.5 µl RNase A for 15 min at 37 °C and with 5 µl proteinase K (Sigma-Aldrich/Merck) for 90 min at 42 °C. DNA was obtained by Phenol:Chloroform:Isoamyl alcohol phase separation and ethanol precipitation and dissolved in 30 µl TE (10 mM Tris-HCl, pH 8.0; 1 mM EDTA).

Chromatin samples were thawed on ice and centrifuged (15 min, 13800 x *g*). Equal amounts of DNA were used for each immunoprecipitation (5-20 µg). Sample volumes were adjusted with dilution buffer (16.7 mM Tris-HCl, pH 8.0; 167 mM NaCl; 1.2 mM EDTA; 0.01% SDS; 1.1% Triton X-100), and 1.5 µl 100 mM PMSF, 1.5 µl PIC and 50 µl protein G-Agarose Salmon Sperm DNA added (protein G-Agarose washed twice with TE, 1 g/ml BSA, 0.1% sodium azide, 0.4 mg/ml Salmon Sperm DNA Beads). Precleaning was done for 2 h on a rotator, antibody added, rotated overnight, 50 µl protein G-Agarose-Salmon Sperm DNA added and rotated for 2 h. Beads were centrifuged, washed by rotation with washing buffer for 10 min, then three times with high-salt wash buffer (50 mM HEPES, pH 7.9; 500 mM NaCl; 1.0 mM EDTA; 0.1% SDS; 1.0% Triton X-100; 0.1% deoxycholic acid sodium) and two times with TE. Beads were resuspended in 300 µl elution buffer (50 mM Tris-HCl, pH 8.0; 10 mM EDTA; 1.0% SDS) and incubated with 20 µl proteinase K (Sigma-Aldrich/Merck) for 2 h at 55 °C and overnight at 65 °C. Beads were removed by centrifugation (5 min, 16000 x *g*) and the supernatant subjected to Phenol:Chloroform:Isoamyl alcohol separation. After ethanol precipitation, the DNA was dissolved in 30 µl TE for PCR. For immunoprecipitation, the following antibodies were used: anti-γH2AX Ser139 (Mouse, monoclonal, Clone JBW 301, 05-636, Merck Millipore), anti-EndoG (Mouse monoclonal, B-2, sc-365359, Santa Cruz Biotechnology, Inc., CA, USA) and IgG (Mouse, sc-2025, Santa Cruz Biotechnology Inc.). Details on immunoprecipitation and Surface plasmon resonance are provided in the Supplementary Methods.

### Immunofluorescence microscopy and Proximity-Ligation-Assay (PLA)

Immunofluorescence microscopy was performed as before[6,30], engaging a BZ-9000 Keyence analyzer and a Nikon 100 × 1.45 oil objective (Keyence, Neu-Isenburg, Germany). The BZ-II Analyzer software was used for quantitative analysis of microscopic images, which were processed by use of ImageJ. Exposure times as well as intensity and minimal focus size thresholds were maintained throughout the whole experimental set.

HeLa cells were cultivated on cover slips in 24-well plates for immunofluorescence microscopy and on Falcon® CultureSlides (Corning Inc., Corning, New York, USA) for PLA, washed three times with PBS for 5 min each, followed by a 1 min pre-extraction step with 0.2% Triton (Sigma-Aldrich/Merck). They were then washed again with PBS for 5 min and fixed with 3.7% formaldehyde (VWR Chemicals BDH) for 10 min.

For immunofluorescence microscopy, the cells were then PBS-washed three times, permeabilized for 12 min with 0.5% 100x Triton/PBS and PBS-washed three times. Blocking was performed with 5% goat serum/PBS for 1 h at RT, followed by a PBS-wash and the primary antibody, diluted in 5% goat serum/PBS, added for 1 h at 37 °C in a humidity chamber. The secondary antibody, diluted in 5% goat serum/PBS, was added after three PBS-washes for 1 h at 37 °C in a humidity chamber in the dark. After three PBS-washes, the nucleus was stained and slides mounted with VectaShield Mounting Medium for Fluorescence containing DAPI (Vector Laboratories, Burlingame, California, USA). The following primary antibodies were used: anti-γH2AX Ser139 (mouse, monoclonal, clone JBW 301, 05-636, Merck Millipore, 1/1000), anti-EndoG (mouse monoclonal, B-2, sc-365359, Santa Cruz Biotechnology Inc., 1/1000) and anti-Ku80 (rabbit monoclonal, S.669.4, Invitrogen/Thermo Fisher Scientific, Waltham, Massachusetts, USA, 1/400). The following primary antibodies were used: goat-anti-mouse Alexa 555 (A21424) or Alexa 488 (A32723), goat-anti-rabbit Alexa 555 (A21428) or Alexa 488 (A11008) from (Invitrogen/Thermo Fisher Scientific, 1/1000).

PLA staining was performed according to the manufacturer's instructions using Duolink® In Situ Detection Reagent Green, Duolink® In Situ PLA Probe Anti-Rabbit PLUS, Duolink® In Situ PLA Probe Anti-Mouse MINUS, and Duolink® In Situ Mounting Medium with DAPI (Sigma-Aldrich/Merck). The primary antibodies (1/300) used were directed against Ku80 (rabbit monoclonal, S.669.4, Invitrogen/Thermo Fisher Scientific), Ku70 (mouse monoclonal, S5C11, sc-56130, Santa Cruz Biotechnology Inc.) and EndoG (mouse monoclonal, B-2, sc-365359, Santa Cruz Biotechnology Inc.). For positive controls, Ku80 and Ku70 were included. Negative controls were performed by including Ku80 or EndoG alone or by omitting antibodies entirely. Images were captured using a Zeiss fluorescence microscope Axio Observer 3/5/7 KMAT (Carl Zeiss AG, Oberkochen, Germany) equipped with appropriate filters for Duolink® In Situ Detection Reagent Green (FITC) and DAPI signals. Image analysis was performed using ZEN Blue software 3.1 (Carl Zeiss AG) with consistent settings maintained across all samples to ensure reproducibility. PLA imaging in Supplementary Fig. 9d was performed using Duolink In Situ Orange Starter Kit Mouse/Rabbit (DUO92102; Sigma) on a BZ-9000 fluorescent microscope and image analysis with BZ-II Analyzer 2.1 software (Keyence, Osaka, Japan).

### Peptide synthesis and nucleofection

The peptides were synthesized and purified by the Core Facility Functional Peptidomics at Ulm University, Ulm Peptide Pharmaceuticals (U-PEP). The peptides were synthesized via solid-phase peptide synthesis (SPPS) as before[30]. Peptides were purified via HPLC-1100 and characterized by mass spectrometry. The lyophilized peptides were stored at 4 °C and after solvation in Aqua ad iniectabilia (B.Braun, Melsungen, Deutschland) at − 20 °C. The peptides are represented by the following aa sequences (see also Table 1): Ku1: ITKEEASGSSV-TAEEAAKKFLAG; Ku2: RVLVKQKKASFEEASNQLINHIEQFL; Ku3:

EIVVQDGITLITKEEASGSSVTAEEAKK; Ku4: ASFEEASNQLINHIEQFLDT-NETPYFMKS; Scrambled: VDSTTIEKSIAEVAEEQESILKKVAGGT; Mut1: EIVVQDGITLITKAEASGSSVTAEEAKK; Mut2: EIVVQDGITLITKEEASGSSV TAEEAAK.

For transfection of cells via Amaxa® nucleofector, called nucleofection, the cells were washed with PBS, re-suspended ($1 \times 10^6$ for K562 and $5 \times 10^5$ for HeLa) in 100 μl Amaxa Cell Line Nucleofector™ Kit V (Lonza, Basel, Switzerland), mixed with 10 μl $H_2O$ as well as peptide and transferred to the cuvettes. The Nucleofector™ II programs used were X-01 for K562 and I-13 for HeLa cells. After nucleofection K562 cells were transferred into 2 ml medium, HeLa cells distributed on three 6-well plates.

## Statistical information

Statistical analyses were performed to evaluate the effects of Ku80-Ct and derived peptides under multiple conditions compared to control (ctrl) groups. Unpaired, nonparametric data were analyzed using the two-tailed Mann-Whitney U test following the Kruskal-Wallis H test. Paired, nonparametric data were analyzed using the Wilcoxon signed-rank test following the Friedman test. Statistical analyses were conducted using GraphPad Prism 8-10 software (GraphPad Software, San Diego, CA, USA). $P$-values < 0.1 are provided, whereby $P$-values < 0.0001 are indicated in the figures by use of asterisks. Data are presented as mean + SD/SEM.

## Single-molecule measurements (SMT)

All methods for SMT, including cell culture, stable cell line generation, cell transduction, transfection, nucleofection and Western blotting, as well as SMT measurements, can be found in the Supplementary Methods.

## Reporting summary

Further information on research design is available in the Nature Portfolio Reporting Summary linked to this article.

## Data availability

All data of this work will be made available upon request without restrictions. Material requests should be addressed to the corresponding author, Lisa Wiesmüller, for questions related to the biomedical part to the corresponding author J. Christof M. Gebhardt, for questions related to SMT and to the modeling part to the corresponding author Elsa Sanchez-Garcia. Source data are provided in this paper.

## Code availability

SMT experiments were analyzed using the TrackIt[40] software code for single-molecule tracking analysis available at https://gitlab.com/GebhardtLab/TrackIt[85].

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

## Acknowledgements

We are grateful to Claus Storgaard Sørensen, Biotech Research and Innovation Center, University of Copenhagen, Denmark, Aswin Mangerich, Nutritional Toxicology, Institute of Nutritional Science, University of Potsdam, Germany, as well as Frank Kirchhoff and Jan Münch, Institute of Molecular Virology, Ulm University, Germany, for enlightening discussions. We would like to express our sincere gratitude to Dierk Niessing, Institute of Pharmaceutical Biotechnology, Ulm University, Germany, for generously providing access to his laboratory equipment and for his valuable consultations in the field of SPR analysis. We thank Jeremy M. Stark, Duarte Cancer Center, CA, for sharing the EJ5SceGFP construct for NHEJ measurements. We thank Ali Omar for his contributions to cloning the monomeric EndoG. SiR HaloTag ligand was kindly provided by Kai Johnsson, Max-Planck-Institute for Medical Research, Heidelberg. We thank the Core Facility FACS of Ulm University for their help with cell sorting, with special thanks to Dr. Simona Ursu, Dr. Sarah Warth, and Daniela Froelich. This project was financially supported by the Deutsche Forschungsgemeinschaft (DFG, German Research Foundation): CRC1279 (no. 316249678), project B05 to L.W. plus J.C.M.G. and project A06 to E.S.-G., CRC1506 (no. 450627322), project B03 to L.W. plus M. R.-S. and project C05 to J.C.M.G., CRC1430 (no. 424228829) project B04 to E.S.-G., Research Grant no. 468578170 to J.C.M.G. as well as under Germany's Excellence Strategy - EXC 2033 – 390677874 – RESOLV to E.S.-G. J.E. was and A.S. is a member of the International Graduate School of Molecular Medicine of the Medical Faculty of Ulm University. E.S.-G was also supported by the DFG.

## Author contributions

L.W., B.G., and J.C.M.G. conceived the study. J.E., A.S., B.G., M.H., and A.R. performed experiments and processed all experimental data. Y.B.R.B. and E.S.G. discovered Ku3 via computational studies. Y.B.R.B. and Y.A.H performed the in silico modeling studies under the supervision of E.S.G; Y.B.R.B., Y.A.H., and E.S.G. designed the computational studies, analyzed and interpreted the results. Y.A.H. and E.S.G. wrote the computational part of the manuscript. J.A.-C. aided in SMT experiments. T.M. designed and T.M. and M.H. performed SPR analysis. L.W., B.G., J.C.M.G., J.E., A.S., M.H., and A.R. jointly designed the experiments, evaluated and interpreted experimental data. M.R.-S. supported experimental studies. L.W., J.C.M.G. and E.S.G. designed the manuscript flow and wrote the initial draft of the manuscript. All authors helped writing the manuscript.

## Funding

## Competing interests

The authors Lisa Wiesmüller, Boris Gole, Julia Eberle (née Wille), J. Christof M. Gebhardt, Anja Reisser (née Palmer), Yasser Bruno Ruiz-Blanco, Elsa Sanchez-Garcia are the inventors of the patent with application number PCT/EP2023/051816; applicant: Ulm University, Germany; status of application: Filed international on January 25, 2023; specific aspect of manuscript covered in patent application: Identification of peptide derived from human Ku80 for use in the prevention of genomic rearrangements causing therapy-related leukemia. The remaining authors declare no competing interests.

## Additional information

¹Department of Obstetrics and Gynecology, Ulm University, Ulm, Germany. ²Department of Physics, Institute of Biophysics, Ulm University, Ulm, Germany. ³Department of Biology, Computational Biochemistry, University of Duisburg-Essen, Essen, Germany. ⁴Department of Biochemical and Chemical Engineering, Chair of Computational Bioengineering, Technical University Dortmund, Dortmund, Germany. ⁵Institute of Pharmaceutical Biotechnology, Ulm University, James-Franck-Ring N27, Ulm, Germany. ⁶Present address: Department of Physics, Institute of Experimental Physics, Ulm University, Ulm, Germany. ⁷Present address: Centre for human molecular genetics and pharmacogenomics, Faculty of Medicine, University of Maribor, SI, Maribor, Slovenia. ⁸Present address: Institut d'Investigacions Biomèdiques August Pi i Sunyer (IDIBAPS), 149-153, Barcelona, Spain. ⁹These authors contributed equally: Julia Eberle, Ahmed Salem, Mara Hofmann, Anja Reisser, Yasser B. Ruiz-Blanco. ✉e-mail: elsa.sanchez@tu-dortmund.de; christof.gebhardt@uni-ulm.de; lisa.wiesmueller@uni-ulm.de

