## [Transparent Peer Review file · Nature Communications]

Discovery of an Endonuclease G-inhibitory Ku80-peptide protecting against leukemogenic rearrangements at the MLL breakpoint cluster

Corresponding Author: Professor Lisa Wiesmüller

Version 0:

Reviewer comments:

Reviewer #1

(Remarks to the Author)

Eberle and colleagues present interesting findings about the role of a peptide Ku80 able to inhibit the enzyme EndoG and protect against leukemogenic rearrangements. The findings are interesting and have potential therapeutic benefit, however, it is felt that more investigations or discussions need to take place about the biological role of this region of Ku80 and its implications in using it as a peptide in treatments. A few points are raised in more detail below.

- Although we appreciate that more experiments could be a lot of work, further biochemical analysis could be carried out to support the binding of Ku80/Ku3 to EndoG. For example, obtaining binding constants for the interaction to quantify how tightly the Ku80 C-terminal domain, and the Ku peptide bind to EndoG. In turn, this will provide insight into the mechanism by which Ku3 interferes with the complex formation between Ku80 + EndoG.

- Further discussions about where this specific region of Ku80 is with regards to recent cryo-EM structures is required. For example, the Ku80-mediated DNA-PK dimer and within the context of Ku70/80. This will in turn give evidence that this region of Ku is not required for other biological mechanisms.

- Although molecular docking was carried out – further structural analysis could be beneficial. Although we appreciate a full structure is a lot of work, AlphaFold with the various constructs may also provide more concrete evidence of the interaction. Minor point.

- In figure 2c, the error bar is high for the control and Ku80-Ct for ChIP- γ H2AX, could the authors comment.

Reviewer #2

(Remarks to the Author)

Reviewer #3

(Remarks to the Author)

This manuscript extends upon the authors' previous reports highlighting the significance of the nuclease EndoG in breakage at the MLL breakpoint cluster region (MLLbcr) by identifying a novel inhibitor-like peptide. The implications of an inhibitor could help expand the therapeutic window for use of doxorubicin for leukaemias, where secondary malignancies through EndoG-induced MLLbcr breakage can be limited. This peptide draws on mimicking *Drosophila* homology, which express an EndoG1 preventing excessive nuclease activity by EndoG that is similar in sequence to the C-terminus of Ku80 (Ku80-ct). The authors conceive a well-ordered workflow, first validating that the Ku80-ct fragment limits EndoG and MLLbcr recombination events, but with no effect on Ku80/DNA-PKcs-dependent DNA repair of doxorubicin-induced breaks. In silico modelling was used to develop an improved, shorter peptide, Ku3, similar in activity to Ku80-ct, and through proximity ligation assays showed that Ku3 competitively disrupts Ku80-EndoG interactions in cells. The work suggests a path to seek inhibitors mitigating the risk of secondary malignancies arising from doxorubicin for treatment of leukaemias. There are,

nonetheless, some substantial technical issues:

Major comments:

1. How much does the overexpression of Ku80, even fragments thereof, mean that there might be an effect on the balance between NHEJ and HR – perhaps influencing recombination effects?
2. The recombination assay raises some concerns. The EGFP⁺ frequency shown in Supp. Fig. 1a shows a handful of cells (0.01%) with a positive result – perhaps this is not an unusual frequency for this particular event but canonical HR assays such as that from the Jasin lab usually gives 1-2% of live cells with positive EGFP⁺, HR⁺ cells.
3. In many instances, there is a lack of a no doxorubicin control to confirm the induction of the EndoG MLLbcr breakage phenomenon. For example, in Fig. 2b, no doxorubicin controls would help confirm that the effects observed aren't an artefact of Ku80-ct overexpression only.
4. The PCR analysis (and essentially all the PCR shown) of the MLLbcr region as a measure of integrity is sound but must be done via quantitative PCR. Band intensity by densitometry is antiquated and inappropriate. And in fact, does not sometimes match up with the representative image – e.g. Fig. 2c, left inset: there are no bands visible in the third lane (Ku80-ct), yet there is not one single point below 50% in the band intensity plot. This also applies to Fig. 5 and Supp. Fig. 5.
5. Fig. 2c: How was the antibody used validated? Off-targeted binding by the antibody is a big risk to the integrity of ChIP data. ChIP-grade antibodies require very stringent validation and this information must be supplied.
6. It is a concern that there is no concentration-dependent effect on the recombination frequency and ultimately the change to recombination frequency in Fig. 4b ($\leq 35\%$) is a modest reduction.
7. It is possible this is a simple mistake but the panel in Supp. Fig. 5 is reported to be a genomic PCR but is, to this reviewer's eye, the result of a separate Western blot. There is a concern regarding data integrity if this is deliberate misleading? Furthermore, why are only two of the four treatments in the right hand inset shown?
8. Could the authors comment on the nature of the Ku80-EndoG interactions in mitochondria, where both doxorubicin-like DNA damage is elevated due to the increased oxidative environment and EndoG expression appears higher? There is some mention of this in the Discussion, but further exploration of this point is needed.

Additional comments:

1. Lines 58-59: The list appears to follow from the cytotoxic drugs conferring sensitivity to MLLbcr breakage as per the sentence prior. The majority of this list include cell contexts and not drugs other than topoisomerase inhibitors – “chromosome loop anchoring...”, “open chromatin...”, “non-B DNA...”.
2. A comment about Ku expression – 0.5 M copies per cell therefore why isn't EndoG under permanent repression in nucleus if MLLbcr breakage is a risk?
3. Fig. 1: needs model of Ku-ct domain and versus EndoG1 motif to illustrate structural relationship.
4. Lines 143-145: The comment about “data scattering precluding detection of significant differences in recombination frequencies” should be omitted or reworded instead to a comment on the unclear effect of the mutant Ku80 and why there may be such variance in the data. Also concerning since data for the same mutant in Supp. Fig. 1 does not show the same variance. It may be best to exclude this mutant within the manuscript entirely or try a Ku80-ct E720/71 mutant.
5. Lines 145-146 and Supp. Fig. 1a: Cell viability through SSC/FSC is an inappropriate measure – could this be improved in a similar FACS experiment with inclusion of a live/dead marker like AO/PI? Also applied to Supp. Fig. 2b.
6. Supp. Fig. 1: For a), it would be nice to also show a representative panel of the presumptive increase of EGFP⁺ cells when the reporter cells were treated with doxorubicin. For b), left panel - representative PI histograms with how the authors have deconvoluted different phases would give confidence to the reader. Right panel – why is the axis broken up to 100% given there are no outliers shown above the 15% mark?
7. Lines 148-156 and Supp. Fig. 2: The I-SceI repair outcome is likely an entirely different HR repair scheme than doxorubicin-induced DNA damage repair. This disparity should be emphasised in the text. It is also inappropriate to use different gates between samples (compare each sample in panel b, autofluorescence-FL2 vs EGFP-FL1 plots).
8. Line 176-177: The sentence beginning “Ectopic expression...” does not include a reference to the figure which it is referring to. This is also the case in Line 266-267.
9. Fig. 2c: If the Ku80 expression levels are similar between ctrl and Ku80-wt, ie in Fig. 2a and Supp. Fig. 3a, then why would recruitment of EndoG not be observed in both cases? Can the authors please comment on this?
10. Line 200: “...did not fully reach statistical significance.” is an odd phrase. Consider rewording.
11. Line 212-213: “...we built a homology model of human EndoG”. Why wasn't AlphaFold used – this is surely now the gold standard?
12. Line 215-217: Why was a sequence-based predictor for protein-protein interactions used instead of an appropriate structural predictor such as Rosetta FlexPepBind?
13. Table 1: can additional peptides be included in this table for comparison?
14. Regarding narrowing down peptide inhibitor, once a small pool was validated in silico, it may have been a possibility to synthesise biotinylated peptide fragments and perform a biotin pulldown of either a whole cell extract or nuclear extract. This would help to confirm both binding to EndoG in vitro or to also establish off-target binders of the inhibitors (e.g. if mass spectrometry is used).
15. Fig. 3: In the bottom inset of panel a, the faded overlay of the Drosophila EndoG1 can be removed for clarity – the reader can compare Ku3 to to the inset above. Can you also include a an axis and rotation to describe the transformation performed between panels a and b?
16. Can evidence of the efficacy of the DNA-PKi in the author's hands be shown? This can be done by monitoring the autophosphorylation of DNA-PKcs at S2056 under DNA damage conditions or verifying enhanced sensitivity to IR upon inhibitor inclusion.
17. Fig 5b and text in reference: Why does the MLLbcr genomic band decrease after 4 h of doxorubicin? Presumably damage/replication-induced recombination rearrangements would not be complete in 4 h, which is too short for most human

HR.

18. Fig. 6: For PLA foci data, the differences between conditions appear small despite reaching significance. It is concerning that for this particular dataset, each cell's foci counts are plotted (increasing n) but in Supp. Fig. 3a, per-cell intensities could have been reported but instead reported the average for each independent experiment. Can the authors explain this choice? It may also be advisable to show single antibody PLA controls ie EndoG and Ku80 alone for each treatment.

19. Referring to line 339-340: Is there not some intrinsic affinity of EndoG for DNA and hence is it really Ku-dependent? Why would inhibited EndoG not be recruited?

20. Lines 322-327: A sequence logo may be a useful means of showing the result of the MSA and to showcase sequence determinants in the optimal EndoG binding motif.

21. While the SMT analysis is interesting, it does not contribute greatly to the progression of the manuscript, especially Fig. 7a-e. The remaining analysis in Fig. 7f onwards is, however, useful and beneficial to the manuscript – though with some caveats (see below).

22. The Halo-EndoG is expressed from the pLV-tetO lentiviral vector. Do the authors have any data on the average expression level of this construct, e.g. by Western blot comparing to WT EndoG? Without endogenous tagging of the WT EndoG, there can be issues with SMT analysis. For example, in what ways might the endogenous EndoG compete with the tagged version, and how much does the overexpression (if it is the case) lead to issues such as aggregation and phase-separated condensates that can affect SMT. Finally, the promoter in these constructs is tetracycline-inducible, yet there is no mention of using doxycycline to relieve inhibition – is expression achieved because there is no TetR transgene in these cells? Can simple IF also be used to show the cellular distribution of EndoG from these constructs – using both the EndoG antibody and HTL-TMR? This might help confirm whether the overexpression leads to subnuclear localisation that differs to endogenous EndoG.

23. There are a few issues in figure legends, e.g. “10 μ M” in the legend of Fig. 2 that should presumably read “10 μ m”.

Reviewer #4

(Remarks to the Author)

In this manuscript, the authors describe a fragment of the Ku80 protein that is able to antagonize the EndoG enzyme and preserve the genome integrity of the MLL major breakpoint region. Starting with the observation that there is sequence similarity between an EndoG inhibitor in fruit flies and the Ku80 c-terminal tail, the team interrogates the sequence to find an optimal peptide for EndoG inhibition in mammals. Following molecular modeling, the team finds 4 fragments to test further with the one named Ku3 having the best ability to prevent instability at MLL while still allowing sensitivity to the Top2 poison doxorubicin. Overall, the idea is to use an EndoG inhibitor to allow for chemotherapy to target cancer cells but not hit fragile regions of the genome that can lead to secondary cancers. Given the significant risk for treatment-related cancers to occur with drugs that induce DNA double-strand breaks, research in this area is significant. Strengths of this work are the molecular modeling, PLA analysis, and single molecule tracking experiments. Major weaknesses are the rigor and significance of the assays monitoring DNA instability. This makes it unclear what the broader impact of this work will be regarding cancer treatment and prevention of secondary malignancy.

1. Figure 1A depicts removal of the DNA-PKcs interaction domain from the Ku80 C-terminus. More recent work from Blundell (PMID: 20023628) and Lees-Miller (PMID: 28641126) indicate the importance of the 592-732 region for DNA-PKcs binding and I'm curious if the authors have been able to completely rule out interaction between Ku1-4 and DNA-PKcs.

2. For experiments in Figure 2 comparing Ku80-wt overexpression to Ku80-ct expression the 3 datapoints are highly inconsistent from experiment to experiment and it is unclear why everything is normalized to Ku80-wt and not the control. With such inconsistencies, more than 3 data points should have been collected. Further, Ku80 levels in most cells is already high and the connection of why Ku80 is bound to the MLL and enhancing EndoG binding is unclear to me. The effect may just be an artifact of Ku80 overexpression. Is the thinking that Ku80 is present at DNA independent of binding to Ku70? The KD of Ku70/80 binding is very low, so there would have to be a stoichiometric imbalance for Ku80 to be present alone. Can the authors overexpress Ku70 to see if the EndoG binding is truly Ku80-dependent?

3. Are breaks at MLL dependent on doxorubicin only or can other Top2 poisons like mitoxantrone also cause breaks there?

4. In experiments shown in Fig 4 combining Ku3 and doxorubicin treatment, the most the recombination frequency decreases is ~35%, would this not mean that a majority of the time breaks at MLL are still occurring to lead to rearrangement? I'm just not sure how effective Ku3 would actually be in the clinic. Also, it is puzzling that at high levels of Ku3 the protective effect is lost.

5. On page 6 line 143, the authors say the experiments with the Ku80 E720A and E721A mutations are inconclusive due to “data scattering”. Is this not just the consequence of the messy nature of FACS analysis that requires further data points? This explanation is used in other part so the manuscript as well. In Fig S4C and S4D, data is concluded with only an n=2, which could impact rigor and reproducibility.

6. The gel assay to monitor MLL stability seems rather low-definition. Are results from the recombination assay, which would only happen if a DSB occurred, not happening at a high enough frequency to detect differences? A quantitative digital PCR assay or even amplicon sequencing of the MLL region may provide a better readout of MLL instability.

7. Has previous work looked genome-wide for regions of Ku80-EndoG binding? This could be an opportunity to determine if other sites in the genome are sensitive to EndoG and if GC content, non-B structures, or DNA methylation is playing a greater role in cleavage.

8. The DNA assay used is homology-based. For Figure S5, what was the recombination frequency, not just the result of the gel assay? An assay that specifically looks at repair through NHEJ would be more appropriate to determine if an inhibitor of DNA-PKcs really has no change in repair frequency in the presence of Ku3 overexpression.

Reviewer #5

(Remarks to the Author)

Version 1:

Reviewer comments:

Reviewer #1

(Remarks to the Author)

The authors have addressed all reviewer comments, and I am happy for the manuscript to be published.

Reviewer #2

(Remarks to the Author)

Reviewer #3

(Remarks to the Author)

The authors have made a substantial effort to address the concerns raised in the previous review and have added a considerable amount of new data. Many of the previously raised issues have been well addressed well. With respect to the major comments, I am satisfied with the authors' responses regarding the use and interpretation of the MLLbcr reporter assay. The explanation for the low recombination frequencies is now clear and appropriate. I also find the discussion of the partial alleviation of MLLbcr breakage by the peptide inhibitor convincing, particularly the authors' acknowledgment that the modest effect size could potentially be improved through the use of combined peptides targeting different EndoG activities. The clarification regarding the apparent PCR bands in Supplementary Fig. 5 are appreciated. The rationale provided for the inability to use qPCR at the MLLbcr is reasonable and adequately discussed in the revised manuscript. That said, it remains somewhat unusual that conventional PCR could therefore be used; it would be helpful if the band densitometry approach were explicitly described in the Methods section, clarifying that MLLbcr band intensities are normalized to appropriate internal control bands in each experiment.

All minor edits have been carefully addressed. I also appreciate that the revised manuscript now clearly states that Ku80-Ct does not affect homologous recombination or non-homologous end joining as measured using I-SceI-based reporter assays (Results, Fig. 1; Supplementary Fig. 2). The inclusion of etoposide as a control (Fig. 1g) is now used consistently to support the conclusion that Ku80-Ct does not affect canonical DSB repair, and the Discussion appropriately reiterates that the effects of Ku80-Ct are independent of both DSB repair pathway choice and DNA-PKcs activity (Discussion, pp. 16–18).

I do have several minor follow-up points related to specific responses:

Regarding Ku80 overexpression, the authors state that there is only limited overexpression of the transgenic Ku80 constructs and refer to Fig. 1e and Fig. 2a. While this is likely correct, it is difficult to fully assess from FLAG-based Western blots alone, as different exposures can strongly influence the apparent degree of overexpression. On re-examination of the source data, it is also unclear why no Ku80-wt-FLAG signal is visible in the FLAG blot shown in "Source Data, Extended Fig. 1: Uncropped Western Blots of Fig. 1e." This raises the question of whether Ku80-Ct may in fact be overexpressed to a much greater extent than Ku80-wt. In addition, the molecular weight markers shown in the uncropped images do not appear to match those in Fig. 1e (e.g., 80/14 kDa versus 70/15 kDa). These issues do not undermine the main conclusions, but addressing them would improve clarity and confidence in the data presentation.

The explanation regarding doxorubicin autofluorescence complicating flow-cytometry-based assays is reasonable. However, I am confused by the apparent lack of EndoG foci induction between mock- and doxorubicin-treated cells in Fig. 2b – though I appreciate that this figure is intended to highlight the EndoG recruitment by ectopic Ku80 expression. In other figures, it is not always clear whether the data represent baseline MLLbcr breakage or doxorubicin-induced effects. For example, in Fig. 2c and 2d, if the data reflect doxorubicin treatment, it would be helpful to explicitly show or reference the induction relative to mock-treated controls.

In the Discussion, the authors speculate that endogenous Ku complex recruits and synergises binding of EndoG to DNA secondary structures at the MLLbcr locus after doxorubicin treatment. Do the authors expect this to apply to other fragile loci?

While the differential chromatin binding of EndoG is well-reported in this study by SMT, I wonder if it could be further validated by analysing chromatin fractions for EndoG in the presence and absence of doxorubicin. This is not essential and could reasonably be left for future work, but a brief comment acknowledging this limitation would be helpful.

Similarly, Ku3 is described as “sufficient to mitigate chemotherapy-related destabilizing effects,” whereas the observed effects are relatively modest. I recommend moderating this wording (e.g., “moderately effective” or “partially mitigates”) to better reflect the magnitude of the observed effects.

Finally, while the authors provide reassurance regarding ChIP antibody validation and convincingly demonstrate EndoG specificity, it remains possible that additional low-abundance bands are present but not visible at the selected exposure. This is a minor point and does not substantially detract from the conclusions, but it is worth acknowledging as a technical limitation.

Reviewer #4

(Remarks to the Author)

I feel the authors have addressed my comments and the new data added to the manuscript improves it. Given the modest reduction that Ku3 expression has in the MLLbcr reporter assay with doxorubicin in Figure 4, and that high concentration mitigates this, it would be interesting if the authors eventually determine if using multiple Ku CTD fragments has a greater impact as they suggest doing in their rebuttal. This may be more convincing in showing this could improve clinical outcomes during chemotherapy treatment. Overall, though, I am supportive of publication in the current form following the revision.

Reviewer #5

(Remarks to the Author)

REVIEWER COMMENTS

Reviewer #1 (Remarks to the Author):

Eberle and colleagues present interesting findings about the role of a peptide Ku80 able to inhibit the enzyme EndoG and protect against leukemogenic rearrangements. The findings are interesting and have potential therapeutic benefit, however, it is felt that more investigations or discussions need to take place about the biological role of this region of Ku80 and its implications in using it as a peptide in treatments. A few points are raised in more detail below.

Reply:

We thank the reviewer for her/his interest in our work and the suggestions that helped to substantially improve our work.

- Although we appreciate that more experiments could be a lot of work, further biochemical analysis could be carried out to support the binding of Ku80/Ku3 to EndoG. For example, obtaining binding constants for the interaction to quantify how tightly the Ku80 C-terminal domain, and the Ku peptide bind to EndoG. In turn, this will provide insight into the mechanism by which Ku3 interferes with the complex formation between Ku80 + EndoG.

Reply:

First, we performed immunoprecipitation (IP) experiments to investigate whether endogenously expressed EndoG and Ku80 can form a complex. Indeed, we were able to detect EndoG coprecipitating with Ku80 both in cells treated with and without aphidicolin, shown to induce replication stress promoting *MLL*bcr cleavage in our previous work (Gole 2015 Oncogene). Note that such co-IPs were not detectable in extracts from doxorubicin-treated cells, whereby the mode-of-action of doxorubicin involving DNA intercalation might play a role.

As suggested, we further performed surface plasmon resonance (SPR) measurements with EndoG and a fragment of Ku80 including the C-terminus of the protein. The measurements revealed strong binding with an equilibrium dissociation constant K_D of 72.7 ± 13.2 nM. In addition, we performed SPR with the Ku3 peptide. However, those measurements failed, supposedly due to the small size of the peptide.

In the revised version of the manuscript we have added co-IP data in the new Supplementary Fig. 1a and the SPR measurement with Ku80 in the new Fig. 1a.

- Further discussions about where this specific region of Ku80 is with regards to recent cryo-EM structures is required. For example, the Ku80-mediated DNA-PK dimer and within the context of Ku70/80. This will in turn give evidence that this region of Ku is not required for other biological mechanisms.

Reply:

The figure below shows a superposition of the EndoG-Ku80_Ct + Ku3 of Fig. 3a, onto Ku80 in the structure of DNAPKcs-Ku80-Ku70-PAXX-XRCC4-DNA ligase 4 (PDB ID: 8BH3; Seif-El-Dahan et al. 2023 Sci Adv, <https://www.science.org/doi/10.1126/sciadv.adg2834>). As depicted, Ku80_Ct (light green in the figure) interacts with both DNA-PKcs and EndoG, while there is no direct interaction between the latter two. The model also shows that EndoG does not interact with Ku70.

On the other hand, the EndoG-Ku3 model shows that, as EndoG forms a symmetric dimer, Ku3 can interact with either protomer of the EndoG dimer. If Ku3 binds to the protomer that interacts with Ku80_Ct (red in the figure), it blocks the direct interaction. If it binds to the other protomer (yellow in the figure), Ku80_Ct could still bind, but, as suggested by the MD simulations, this binding induces an increase in the EndoG interchain dimer flexibility (Supplementary Fig. 6c), which could destabilize the interaction with EndoG.

- Although molecular docking was carried out – further structural analysis could be beneficial. Although we appreciate a full structure is a lot of work, AlphaFold with the various constructs may also provide more concrete evidence of the interaction.

Reply:

Although AlphaFold (AF2, AF3) is state of the art for structural prediction, it is not the method of choice when the dynamics of the system is important. In that case, molecular dynamics simulations are needed. Furthermore, AlphaFold cannot model a system of the size of PDB ID: 8BH3. AF2 has a hard limit of 5000 residues, while the publicly available model AF3 has a 2048 residues limit. The 8BH3 structure has 6920 (x2) residues.

We performed extensive and unbiased MD simulations to show the dynamic behaviour of the EndoG-Ku3 complex, that complements the molecular docking prediction. We identified several binding modes, one of which largely resembles the one predicted with molecular docking. Hence, the structural characterization performed with the experimental structures available, molecular docking, and MD simulations is more comprehensive than any AF model.

Minor point

- In figure 2c, the error bar is high for the control and Ku80-Ct for ChIP- γ H2AX, could the authors comment.

Reply:

Very different from transcription or chromatin factors, only a small fraction of EndoG molecules enter the nucleus, as only limited introduction of chromosomal breaks is compatible with life (Gole et al. 2018 Leukemia; Liu et al. 2016 Mol Cell Oncol; new Supplementary Fig. 10a). Therefore, *MLLbcr*-specific ChIP analyses of EndoG and of γ H2AX at the *MLLbcr*, marking DNA double-strand breaks (DSBs) after local EndoG-mediated cleavage, are challenging, which explains larger standard deviations (SDs). To address the larger SD seen for γ H2AX, which is caused by one single experiment showing a smaller effect, we newly added data from one more experiment (Fig. 2c). Still, please note that each data point for the ctrl and Ku80-Ct expressing cells is below the corresponding one for Ku80-wt expression, i.e. each experiment shows the same trend. Finally, please note that the ChIP values for Ku80-Ct versus Ku80-wt decrease in Figure 2c even though the input PCR using the same primers increases in Fig. 2d, indicating an underestimation rather than overestimation of the observed effect of Ku80-Ct during ChIP experiments.

Reviewer #3 (Remarks to the Author):

This manuscript extends upon the authors' previous reports highlighting the significance of the nuclease EndoG in breakage at the MLL breakpoint cluster region (MLLbcr) by identifying a novel inhibitor-like peptide. The implications of an inhibitor could help expand the therapeutic window for use of doxorubicin for leukaemias, where secondary malignancies through EndoG-induced MLLbcr breakage can be limited. This peptide draws on mimicking *Drosophila* homology, which express an EndoGI preventing excessive nuclease activity by EndoG that is similar in sequence to the C-terminus of Ku80 (Ku80-ct). The authors conceive a well-ordered workflow, first validating that the Ku80-ct fragment limits EndoG and MLLbcr recombination events, but with no effect on Ku80/DNA-PKcs-dependent DNA repair of doxorubicin-induced breaks. In silico modelling was used to develop an improved, shorter peptide, Ku3, similar in activity to Ku80-ct, and through proximity ligation assays showed that Ku3 competitively disrupts Ku80-EndoG interactions in cells. The work suggests a path to seek inhibitors mitigating the risk of secondary malignancies arising from doxorubicin for treatment of leukaemias. There are, nonetheless, some substantial technical issues:

Reply:

We are truly thankful for the diligent evaluation and the constructive comments made by this expert.

Major comments:

1. How much does the overexpression of Ku80, even fragments thereof, mean that there might be an effect on the balance between NHEJ and HR – perhaps influencing recombination effects?

Reply:

As demonstrated by the Western blots of total Ku80 levels before and after expression of exogenous Ku80 in Fig. 1e and 2a we do not overexpress exogenous Ku80 excessively.

Since we also did not see an effect of Ku80-Ct on homologous DSB repair after targeted cleavage of the corresponding reporter construct (Supplementary Fig. 2b), we conclude that Ku80-Ct affects *MLLbcr* rearrangements independently of a direct or indirect effect on a particular DSB repair activity and rather is active upstream of DSB repair, namely – as an endonuclease - during formation of the break.

This argument is strengthened by the fact that we do not see a significant effect of Ku80-Ct on etoposide-induced *MLLbcr* rearrangements (see new Fig. 1g), which through topoisomerase II-poisoning complex formation directly generates DSBs, while the mode-of-action of doxorubicin relies on DNA adduct formation and DNA intercalation and thereby formation of R-loops, ROS-induced replication stress and DNA-breaks (Yang 2014 BBA).

To further address the comment of reviewer #3, we newly performed measurements of NHEJ for mock-transfected (ctrl), Ku80-wt and Ku80-Ct expressing K562 cells. As with homologous DSB repair, NHEJ frequencies did not significantly differ between ctrl, Ku80-wt and Ku80-Ct expressing cells.

2. The recombination assay raises some concerns. The EGFP+ frequency shown in Supp. Fig. 1a shows a handful of cells (0.01%) with a positive result – perhaps this is not an unusual frequency for this particular event but canonical HR assays such as that from the Jasin lab usually gives 1-2% of live cells with positive EGFP+, HR+ cells.

Reply:

Please note that “canonical HR assays” rely on targeted cleavage of the reporter construct by an overexpressed endonuclease (e.g. I-SceI), so that most copies of the reporter DNA in the majority of cells are subject to DSB repair (transfection efficiencies at 70-80% in our study). We used such “canonical” I-SceI-mediated repair assays in Supplementary Fig. 2 resulting in repair frequencies of 0.3-4.4% (EGFP+ cells among living and transfected cells), which lies in the range of repair frequencies of 0.05-5% seen by our colleagues (Pierce and Jasin 1999 Genes Dev; Bennardo et al. 2008 Plos Genet). In our work we use the same reporter plasmid in case of the I-SceI- and doxorubicin-induced assays except for the insertion of the *MLLbcr* spacer in the latter. Doxorubicin induces breaks at the *MLLbcr* spacer, namely indirectly by endogenous enzymes at sites of replication stress/secondary DNA structures, during repair of oxidative damage and/or during adduct removal as well as during release of torsional stress (Yang 2014 BBA; Gole and Wiesmüller 2015 Front Cell Dev Biol). However, such breaks are obviously generated at a much lower frequency during a doxorubicin pulse (4 h) than by targeted and repeated cleavage during the whole period of I-SceI expression between transfection and harvest of the cells (48 h).

Still, EGFP+ cells, e.g. 99 in Supplementary Fig. 1b, were detected faithfully by FACS, with similar gating strategies and with similar numbers of EGFP+ cells as seen by users of the Jasin assay (see e.g. Pierce and Jasin 1999 Genes Dev; Bennardo et al. 2008 Plos Genet). Note that the reporter constructs used by us and others are comparable (Pierce and Jasin 1999 Genes Dev; Akyüz 2002 Nucleic Acids Res; Bennardo et al. 2008 Plos Genet). Importantly, the sensitivity of these assays is determined by the total number of living cells analyzed by FACS. For chemotherapeutic treatment-induced recombination at the chromosomally integrated *MLLbcr* reporter we typically analyze 0.5-1 million viable cells to compensate for the low recombination frequency of around 0.01%. However, for the I-SceI induced, i.e. maximally efficient DSB repair, typically 10 000-50 000 viable cells are analyzed by FACS (Pierce and Jasin 1999 Genes Dev; Bennardo et al. 2008 Plos Genet; this work), ultimately yielding similar counts of EGFP+ cells.

3. In many instances, there is a lack of a no doxorubicin control to confirm the induction of the EndoG *MLLbcr* breakage phenomenon. For example, in Fig. 2b, no doxorubicin controls would help confirm that the effects observed aren't an artefact of Ku80-ct overexpression only.

Reply:

In the revised version of the manuscript, Fig. 2b, we show the requested EndoG foci data without doxorubicin. Moreover, we added such data from mock-treatments for *MLLbcr* rearrangements in Fig. 1g (together with viabilities in Supplementary Fig. 1d), for nuclear Ku80 intensities in Supplementary Fig. 3a, PLA data in Supplementary Fig. 9a-c and genomic PCR data in Fig. 5c. Please note that for recombination measurements of doxorubicin-treated cells a specific autofluorescence-FL2/EGFP-FL1 gate was used due to the red autofluorescence of doxorubicin (see Supplementary Fig. 1b). Therefore mock-treated cells were compared with etoposide-treated cells in Fig. 1g to illustrate treatment-induced recombination augmentation.

4. The PCR analysis (and essentially all the PCR shown) of the *MLLbcr* region as a measure of integrity is sound but must be done via quantitative PCR. Band intensity by densitometry is antiquated and inappropriate. And in fact, does not sometimes match up with the representative image – e.g. Fig. 2c, left inset: there are no bands visible in the third lane (Ku80-ct), yet there is not one single point below 50% in the band intensity plot. This also applies to Fig. 5 and Supp. Fig. 5.

Reply:

We unfortunately had to apply conventional quantification of band intensities after PCR amplifications because of the proneness of the *MLLbcr* to form extended secondary structures (exacerbated by doxorubicin treatment, Yang 2014 BBA), which are known to cause problems especially during qPCR (see e.g. Fan 2019 J Biomol Struct Dyn 2019). We encountered these problems during method establishment, when also exploring the possibility of using qPCR, regardless of primer target sites, additives and other optimization measures. Importantly, PCR band intensities were only quantified in the linear range according to our ChemiDoc™ MP Imaging System (Bio-Rad Laboratories, Hercules, California, USA). Moreover, we routinely included 50% template controls in our PCR runs (now shown in the revised manuscript in Fig. 5c and Supplementary Fig. 5a).

We agree with reviewer #3 that the blot/exposure shown in Fig. 2c, left panel for ChIP-EndoG, was not representative, why we replaced it by a more representative one from a second PCR amplification of the same sample. Regarding values below 50%, we would like to bring to attention that in the corresponding ChIP-EndoG graph indeed there is one value below 50% for Ku80-Ct (see also Source Data). More importantly, each data point for Ku80-Ct shows a decrease of the band intensity compared to Ku80-wt, i.e. shows the same trend. Please also note that the ChIP values for Ku80-Ct versus Ku80-wt decrease in Fig. 2c even though the input PCR using the same primers increases in Fig. 2d, indicating an underestimation rather than overestimation of the observed effect of Ku80-Ct during ChIP experiments.

5. Fig. 2c: How was the antibody used validated? Off-targeted binding by the antibody is a big risk to the integrity of ChIP data. ChIP-grade antibodies require very stringent validation and this information must be supplied.

Reply:

Indeed, our ChIP antibodies directed against EndoG and γ H2AX were carefully validated. First, we chose monoclonal antibodies to minimize batch effects. Second, already in our previous work Gole 2018 Leukemia, we ensured disappearance of the specific EndoG band upon knockdown. Third, we compared different antibodies verifying absence of non-specific bands (see also new anti-EndoG immunoblot in Supplementary Fig. 1a and uncropped image in Source Data). Fourth, we newly generated a Western blot (see Supplementary Fig. 3b and Source Data), where we demonstrate increased γ H2AX signals after DSB-inducing treatment.

6. It is a concern that there is no concentration-dependent effect on the recombination frequency and ultimately the change to recombination frequency in Fig. 4b ($\leq 35\%$) is a modest reduction.

Reply:

We also noticed that downregulation of *MLLbcr* rearrangements by the Ku3 peptide was more modest than by Ku80-Ct, which caused a reduction of the recombination frequency down to 52% of the frequency measured with Ku80-wt. Given that our EndoG-Ku3 binding simulations suggest that peptide Ku3 interferes with EndoG-Ku80 complex formation and modulates chromatin binding of EndoG dimers, there is a possibility that these two types of interference differentially affect endo- and exonucleolytic activities of EndoG dimers and monomers, respectively. Furthermore, Ku80-Ct interacts with EndoG through multiple regions along its sequence clustering around residues 686-707 and to a lesser extent also around residues 599-624 (Fig. 4a).

Therefore, in the discussion section on page 19 we propose that combining peptides may optimize binding affinities and specificities.

7. It is possible this is a simple mistake but the panel in Supp. Fig. 5 is reported to be a genomic PCR but is, to this reviewer's eye, the result of a separate Western blot. There is a concern regarding data integrity if this is deliberate misleading? Furthermore, why are only two of the four treatments in the right hand inset shown?

Reply:

We assure that the images presented in the original and revised manuscript versions are all depicting genomic PCR bands in an agarose gel (Fig.s 2b,c, 5b,c; Supplementary Fig. 5a). The confusion may stem from the fact that in Fig. 2 and 5a we used SeaKem® LE Agarose (Lonza, Basel, Switzerland) for separation of DNA fragments, whereas in Fig. 5c and Supplementary Fig. 5 we made the effort to improve the resolution of the electrophoretic separation by use of UltraPure™ Agarose-1000 (Invitrogen/Thermo Fisher Scientific). These details are mentioned in the corresponding figure legends of the revised manuscript. In the revised version of the manuscript we further present a representative image of the agarose gel for each treatment.

8. Could the authors comment on the nature of the Ku80-EndoG interactions in mitochondria, where both doxorubicin-like DNA damage is elevated due to the increased oxidative environment and EndoG expression appears higher? There is some mention of this in the Discussion, but further exploration of this point is needed.

Reply:

As hopefully better explained in the discussion section on page 18 of the revised manuscript, there is a need to protect the nuclear genome from excessive and unspecific nucleolytic activity of human EndoG. Even under sublethal conditions, when only a small fraction of EndoG is detectable in the nucleus, EndoG must be tightly controlled to prevent chromosomal instability (Liu 2016 Mol Oncol; Hawkins 2021 IJMS). According to our model the Ku complex, through physical interactions of EndoG with the Ku80 C-terminal domain, synergistically aids localization of EndoG to specific secondary structures in the genome to support resolution of these replication barriers.

In mitochondria of unstressed cells EndoG resides in the intermembrane space, i.e. is spatially separated from the mitochondrial genome (Ohsato 2002 Eur J Biochem). Therefore, such a control mechanism may not be necessary in mitochondria. This can also explain why an alternate, C-terminally truncated form of Ku80 is localized in mammalian mitochondria (Coffey and Campbell 2000 Nucleic Acids Res). Under conditions of oxidative stress, our recent works revealed a role of EndoG in nucleolytic removal of irreparably oxidatively damaged copies, whereby multiple remaining intact copies of the mitochondrial genome and compensatory mitochondrial DNA replication ensure mitochondrial genome homeostasis (Wiehe 2018 Oncotarget; Schreier 2022 Genes). Altogether, while the nucleolytic activity of human EndoG in the nucleus must be tightly controlled to prevent chromosomal instability, proposedly through interactions with the EndoG-like Ku80 C-terminal domain, such control mechanism seems to be obsolete in mitochondria.

Additional comments:

1. Lines 58-59: The list appears to follow from the cytotoxic drugs conferring sensitivity to MLLbcr breakage as per the sentence prior. The majority of this list include cell contexts and not drugs other than topoisomerase inhibitors – “chromosome loop anchoring...”, “open chromatin...”, “non-B DNA...”.

Reply:

We thank reviewer #3 for careful reading and amended these two sentences as follows:

“Compilation of breakpoint distribution data from patients together with experimental results have connected clustered breakage at the *MLLbcr* with sensitivity to various cytotoxic drugs, such as topoisomerase inhibitors. Additional cell context-dependent features destabilizing the *MLLbcr* range from chromosome loop anchoring by CTCF and Cohesin, open chromatin (DNase I hypersensitivity), formation of a complex non-B DNA secondary structure predisposing genomic regions to breakage, RNA Polymerase II binding, transcription and R-loop formation to replication stress as well as apoptotic cleavage.”

2. A comment about Ku expression – 0.5 M copies per cell therefore why isn't EndoG under permanent repression in nucleus if MLLbcr breakage is a risk?

Reply:

Our model implies Ku complex-mediated recruitment of EndoG to specific sites in the chromatin, where Ku70, Ku80 and EndoG synergistically recognize energetically stable secondary structures such as predicted for the *MLLbcr* (Gole and Wiesmüller 2015 Front Cell Dev Biol) and augment resolution of these replication barriers by EndoG. Competition between exogenous Ku80-Ct and endogenous Ku80 for EndoG binding reduces the long bound fraction of EndoG, and therefore presumably the activity of EndoG on DNA. Ku3 interactions with EndoG are competitive with regard to EndoG-Ku80 complex formation, as predicted by the *in silico* molecular modeling and confirmed with the PLA experiments. Additionally, these interactions can modulate the flexibility of the EndoG dimer, affecting the EndoG-Ku80 complex formation, as suggested by the molecular dynamics simulations.

We explain this model on page 17 of the revised version of the manuscript.

3. Fig. 1: needs model of Ku-ct domain and versus EndoGI motif to illustrate structural relationship.

Reply:

Fig. 1, now also including the SPR data, is already displaying a number of panels, which are key for the reader to understand the concept and initial methodologies of the manuscript, so that we prefer to leave the structural modeling in Fig. 3.

4. Lines 143-145: The comment about “data scattering precluding detection of significant differences in recombination frequencies” should be omitted or reworded instead to a comment on the unclear effect of the mutant Ku80 and why there may be such variance in the data. Also concerning since data for the same mutant in Supp. Fig. 1 does not show the same variance. It may be best to exclude this mutant within the manuscript entirely or try a Ku80-ct E720/71 mutant.

Reply:

As suggested by reviewer #3 we excluded data obtained for mutant Ku80(E720A, E721A) from the manuscript.

5. Lines 145-146 and Supp. Fig. 1a: Cell viability through SSC/FSC is an inappropriate measure – could this be improved in a similar FACS experiment with inclusion of a live/dead marker like AO/PI? Also applied to Supp. Fig. 2b.

Reply:

Please note that we assessed viability under the conditions of recombination measurements in Fig. 1d not only by SSC/FSC gating but additionally by sub-G1 DNA content analysis of propidium iodide (PI) stained K562 reporter cells. These data are depicted in Supplementary Fig. 1b and Supplementary Fig. 1c. Moreover, we performed sub-G1 analysis of HeLa cells under the conditions of ChIP/genomic PCR and immunofluorescence microscopy (Fig. 2; Supplementary Fig. 3a,b) in Supplementary Fig. 3c. Also, pan-nuclear γ H2AX analyses of HeLa cells (Supplementary Figs. 3b) did not provide evidence for altered doxorubicin-induced toxicities by Ku80-Ct. To provide an additional measure of cell viabilities, we performed MTT assays with doxorubicin treated HeLa cells expressing Ku80-wt or K80-Ct as compared to mock-transfected cells (see new Supplementary Fig. 3d).

6. Supp. Fig. 1: For a), it would be nice to also show a representative panel of the presumptive increase of EGFP+ cells when the reporter cells were treated with doxorubicin. For b), left panel - representative PI histograms with how the authors have deconvoluted different phases would give confidence to the reader. Right panel – why is the axis broken up to 100% given there are no outliers shown above the 15% mark?

Reply:

Please note that for recombination measurements with doxorubicin-treated cells a special autofluorescence-FL2/EGFP-FL1 gate must be applied due to the red autofluorescence of doxorubicin (now explicitly mentioned in the legend to Supplementary Fig. 1b). Therefore, we prefer not to directly compare recombination measurements in mock- and doxorubicin-treated cells. Accordingly, we also do not present recombination frequencies of mock-treated cells in the corresponding Fig. 1f.

In the revised version, we additionally show recombination data, viabilities and the corresponding FACS-plots of mock- and etoposide-treated cells in the new Fig. 1g and new Supplementary Fig. 1d. Though we fully separate recombination data from mock- and doxorubicin-treated cells in the revised version of the manuscript, on average we find an increase in the percentage of living EGFP+ cells in doxorubicin- as compared to mock-treated cells (see legends to Fig. 1f and g).

As requested we illustrate our gating strategies for determination of cells in different cell cycle stages and for identification of cells with sub-G1 content by representative histograms of PI stained cells (FL2) on the right hand side of Supplementary Fig. 1c.

As suggested we present the sub-G1 data in the lower panel of Supplementary Fig. 1c with a maximum value of 15% at the Y-axis in the revised version of the manuscript.

7. Lines 148-156 and Supp. Fig. 2: The I-SceI repair outcome is likely an entirely different HR repair scheme than doxorubicin-induced DNA damage repair. This disparity should be emphasised in the text. It is also inappropriate to use different gates between samples (compare each sample in panel b, autofluorescence-FL2 vs EGFP-FL1 plots).

Reply:

Regarding the repair outcome in I-SceI expressing and doxorubicin-treated cells we agree with reviewer #3 that the trigger for repair might be different in the two scenarios. While I-SceI generates a DSB in the reporter (within the 5' positioned mutated *EGFP* gene), doxorubicin intercalates into DNA thereby promoting formation of R-loops, ROS-induced replication stress and DNA-breaks (Yang 2014 BBA) clustering in the *MLLbcr* spacer of the reporter (Ireno 2014 Arch Toxicol; Eberle 2021 Front Oncol). To account for reviewer #3's comment, we measured *MLLbcr* rearrangements in K562 reporter cells replacing doxorubicin by etoposide, known to generate DSBs in the *MLLbcr* spacer through topoisomerase II-poisoning (Yang, Teves 2014 BBA; Gole 2015 Oncogene). Though recombination frequencies were stimulated by etoposide- versus mock-treatment, Ku80-Ct neither had an effect on recombination (new Fig. 1g) nor on viabilities (new Supplementary Fig. 1d). In the same paragraph on page 7 we emphasize that repair of DSBs generated by targeted cleavage might be different from repair of doxorubicin-induced breaks and explain the different modes-of-action of doxorubicin and etoposide.

We apologize for the mistake of having presented the data of the second panel in the original version of Supplementary Fig. 2 with a different gate for EGFP+ cells. Of course, we used the same gates for all the samples throughout each experiment except for doxorubicin-treated cells due to the red autofluorescence of doxorubicin (see reply to additional comment 6. above). Accordingly, we replaced the representative FACS plots in Supplementary Fig. 2b of the revised version of the manuscript by new ones.

8. Line 176-177: The sentence beginning "Ectopic expression..." does not include a reference to the figure which it is referring to. This is also the case in Line 266-267.

Reply:

We added the reference to the Figure in both cases.

9. Fig. 2c: If the Ku80 expression levels are similar between ctrl and Ku80-wt, ie in Fig. 2a and Supp. Fig. 3a, then why would recruitment of EndoG not be observed in both cases? Can the authors please comment on this?

Reply:

Indeed, quantifying and normalizing band intensities in Western blots of total Ku80 from HeLa cells with and without expression of exogenous Ku80 also did not reveal significant differences. Presentation of microscopic Ku80 signals as per cell quantification values, as proposed by Referee #3 (see reply to additional comment 18. ad Supplementary Fig. 3a), even suggests a decrease of nuclear Ku80 signals in cells expressing exogenous Ku80-wt.

We can only speculate about different features of pre-existing Ku80, stabilized by Ku70 heterodimerization (Koike 2002 J Radiat Res), and *de novo* produced Ku80, such as regarding

differential post-translational modifications that might impact on nuclear localization and chromatin binding.

10. Line 200: "...did not fully reach statistical significance." is an odd phrase. Consider rewording.

Reply:

We deleted this sentence, as p-values are already indicated in the figure itself.

11. Line 212-213: "...we built a homology model of human EndoG". Why wasn't AlphaFold used – this is surely now the gold standard?

Reply:

As already discussed before, despite AlphaFold (AF2, AF3) being considered as the state of the art for structural prediction, in cases when the dynamics of the system is important, molecular dynamics is the method of choice. Additionally, the agreement between modelling and the experimental results evidences that homology modelling is still a valid and accurate technique.

In this case, several structures of the Ku80+Ku70+DNA-PKcs complex experimentally available deliver sufficient information to interpret and model further interactions with this complex, without employing AlphaFold. Furthermore, for modelling EndoG-Ku3, where the flexibility and dynamics of the complex are key features, we performed extensive and unbiased MD simulations to show the dynamic behaviour of the EndoG-Ku3 complex, complementing the molecular docking prediction. We identified several binding modes, one of which largely resembles the one predicted with molecular docking. Hence, the structural characterization performed with the experimental structures available, molecular docking, and MD simulations is more comprehensive than any AF model.

12. Line 215-217: Why was a sequence-based predictor for protein-protein interactions used instead of an appropriate structural predictor such as Rosetta FlexPepBind?

Reply:

Given the high number of sequences to be screened (100000), we chose our *in-house* sequence-based method, PPI-Detect (a machine learning-based method to predict the likelihood of interaction of protein-protein sequences, see Kling et al. 2023 Int J Mol Sci; Romero-Molina et al. 2019 J Comput Chem), which can perform the screening in a fraction of the time that Rosetta's FlexPepBind could do it.

Additionally, because the PPI-Detect method is trained on interacting and non-interacting sequences, it does not suffer from pitfalls that can affect structure-based methods, like insufficient quality of the structures or insufficient sampling of the flexibility. Furthermore, in the screening step where PPI-Detect was used, we were interested in predicting whether or not the peptide sequence would interact with EndoG, instead of the actual tridimensional structure of the complexes. We then performed the docking simulations on the filtered sequences. We would like to note that our sequence-based method was trained with a big, non-redundant dataset that includes information about experimentally proven interacting and non-interacting domains. This information was later used to generate molecular descriptors of the residues that codify for the interaction of a pair of sequences. Therefore, even if PPI-Detect is a sequence-based predictor, it

benefits from the structural information encoded within the method thanks to the generated descriptors.

Due to all of the above, PPI-Detect is a more suitable tool for massive screening of thousands of sequences than FlexPepBind.

13. Table 1: can additional peptides be included in this table for comparison?

Reply:

Table 1 shows all the peptides experimentally tested. Given the promising results found for Ku3 as lead compound, no more peptides were experimentally tested, as the goal of the in-silico exploration was precisely to reduce experimental costs by identifying potential lead compounds.

14. Regarding narrowing down peptide inhibitor, once a small pool was validated in silico, it may have been a possibility to synthesise biotinylated peptide fragments and perform a biotin pulldown of either a whole cell extract or nuclear extract. This would help to confirm both binding to EndoG in vitro or to also establish off-target binders of the inhibitors (e.g. if mass spectrometry is used).

Reply:

To consider the thoughts of referee #1, we performed immunoprecipitation (IP) experiments to investigate whether endogenously expressed EndoG and Ku80 can form a complex. Indeed, we were able to detect EndoG coprecipitating with Ku80 both in cells treated with and without aphidicolin, shown to induce replication stress promoting *MLL*bcr cleavage in our previous work (Gole 2015 Oncogene). Note that such co-IPs were not detectable in extracts from doxorubicin-treated cells, whereby the mode-of-action of doxorubicin involving DNA intercalation might play a role.

We further performed surface plasmon resonance (SPR) measurements with EndoG and a C-terminal fragment of Ku80. The measurements revealed strong binding with an equilibrium dissociation constant KD of 72.7 ± 13.2 nM. In addition, we performed SPR with the Ku3 peptide. However, those measurements failed, supposedly due to the small size of the peptide.

In the revised version of the manuscript we have added co-IP data in the new Supplementary Fig.1a and the SPR measurement with Ku80 in the new Fig. 1a.

We also attempted to perform PLA for EndoG and tagged Ku3, which however failed due to insufficient anti-tag antibody specificities.

In sum, we provide biochemical evidence for EndoG-Ku80 complex formation in living cells and direct EndoG-Ku80-Ct interactions *in vitro*. Biochemical analysis of EndoG-Ku3 peptide complex formation is more challenging and will have to await future investigations.

15. Fig. 3: In the bottom inset of panel a, the faded overlay of the Drosophila EndoGI can be removed for clarity – the reader can compare Ku3 to the inset above. Can you also include an axis and rotation to describe the transformation performed between panels a and b?

Reply:

The overlay was removed and two rotation axes were included.

16. Can evidence of the efficacy of the DNA-PKi in the author's hands be shown? This can be done by monitoring the autophosphorylation of DNA-PKcs at S2056 under DNA damage conditions or verifying enhanced sensitivity to IR upon inhibitor inclusion.

Reply:

As proposed, we performed Western blotting experiments to monitor autophosphorylation of DNA-PKcs at S2056 after a 4 h treatment with etoposide with and without DNA-PKi (NU7441). Hereby, we confirm the efficacy of DNA-PKcs inhibition, as shown in Supplementary Fig. 5b, right panel.

17. Fig 5b and text in reference: Why does the *MLLbcr* genomic band decrease after 4 h of doxorubicin? Presumably damage/replication-induced recombination rearrangements would not be complete in 4 h, which is too short for most human HR.

Reply:

Please note that genomic PCR at the *MLLbcr* is monitoring loss of the integrity of the *MLLbcr* such as through breakage during a 4 h doxorubicin pulse (Gole 2015 Oncogene; Gole 2018 Leukemia). Subsequent recombination at the *MLLbcr*, which is monitored by the reporter-based assay, indeed takes longer why recombination measurements were performed 72 h after the doxorubicin pulse. Since the mode-of-action of doxorubicin relies on DNA adduct formation and DNA intercalation and thereby formation of R-loops, ROS-induced replication stress and ultimately DNA-breaks (Yang 2014 BBA), slower repair processes relying on end processing like recombination are involved as compared to fast canonical NHEJ religating clean DSBs (Löbrich and Jeggo 2017 Trends Biochem Sci).

18. Fig. 6: For PLA foci data, the differences between conditions appear small despite reaching significance. It is concerning that for this particular dataset, each cell's foci counts are plotted (increasing n) but in Supp. Fig. 3a, per-cell intensities could have been reported but instead reported the average for each independent experiment. Can the authors explain this choice? It may also be advisable to show single antibody PLA controls ie EndoG and Ku80 alone for each treatment.

Reply:

We thank reviewer #3 for this attentive comment and present nuclear Ku80 data in Suppl. Fig. 3a as per cell quantification values consistent with the matching EndoG foci counts in Fig. 2b and PLA counts in Fig. 6. Moreover, we newly provide PLA data in mock-treated cells (Supplementary Fig. 9a-b), revealing the same pattern of reduction of Ku80-EndoG complex formation by Ku3 but not by scrambled Ku3, mutated Ku3 (Ku3-Mut1, Ku3-Mut2) or Ku1 at the lower PLA foci level of untreated cells (Fig. 6a). These observations support the concept of replication-associated action of Ku80-EndoG complexes on the *MLLbcr* and possibly related sequences under sublethal conditions (Gole 2015 Oncogene; Gole 2018 Leukemia) that can be provoked by doxorubicin-inducing treatment. Indeed, we show in the new Supplementary Fig. 9d that enforced replication stress by aphidicolin treatment augments association of Ku80 and EndoG according to PLA.

Finally, we provide quantifications of single antibody PLA controls in Supplementary Fig. 9b, showing that these background signals are at the level of Ku3-repressed Ku80-EndoG PLA foci/nucleus.

19. Referring to line 339-340: Is there not some intrinsic affinity of EndoG for DNA and hence is it really Ku-dependent? Why would inhibited EndoG not be recruited?

Reply:

Being a nuclease, EndoG for sure has affinity for DNA, but as with all DNA-binding complexes individually weak DNA interactions can sum up to a synergistic effect. In analogy, transcriptional activator synergy is based on cooperative DNA binding (Langdon & Hochschild 1999 PNAS). Inhibition of Ku80-Endo G complex formation by Ku3 will prevent such synergistic effect by Ku80, Ku70 and EndoG, which all three are known to recognize G4 and other non B-DNA secondary structures in the genome such as cruciforms or hairpins (Postow 2011 FEBS Lett; Zhang 2012 Anal Bioanal Chem; Zanden 2020 Nucleic Acids Res; Dahal 2022 Elife).

20. Lines 322-327: A sequence logo may be a useful means of showing the result of the MSA and to showcase sequence determinants in the optimal EndoG binding motif.

Reply:

Below is provided a sequence logo generated with the WebLogo server 3.7.9 (<https://weblogo.threeplusone.com/>, Crooks et al. 2004 Genome Res; Schneider et al. 1990 Nucleic Acids Res). We used the multiple sequence alignment (MSA) of the peptides generated to identify the conserved sequence motif that interacts with EndoG. The figure is now Supplementary Fig. 7.

Supplementary Fig. 7. Weblogo of the multiple sequence alignment (MSA) of 18960 peptides with an interaction score with EndoG equal to or larger than that of Ku3. The letters represent the conservation of the residues at the given position. Empty positions represent a gap in the MSA. Small letters represent non-conserved positions in the MSA.

21. While the SMT analysis is interesting, it does not contribute greatly to the progression of the manuscript, especially Fig. 7a-e. The remaining analysis in Fig. 7f onwards is, however, useful and beneficial to the manuscript – though with some caveats (see below).

Reply:

Since single-molecule tracking is a relatively new method and not all readers might be familiar with the appearance of raw data and the subsequent analysis steps, we think it is important to show both measurement schemes (Fig. 7 a, c), raw data (Fig. 7 a,d) and intermediate steps of the analysis pipeline (Fig. 7 b, e, f) in the main manuscript and not only as supplementary material, before showing the final results (Fig. g, h).

22. The Halo-EndoG is expressed from the pLV-tetO lentiviral vector. Do the authors have any data on the average expression level of this construct, e.g. by Western blot comparing to WT EndoG? Without endogenous tagging of the WT EndoG, there can be issues with SMT analysis. For example, in what ways might the endogenous EndoG compete with the tagged version, and how much does the overexpression (if it is the case) lead to issues such as aggregation and phase-separated condensates that can affect SMT. Finally, the promoter in these constructs is tetracycline-inducible, yet there is no mention of using doxycycline to relieve inhibition – is expression achieved because there is no TetR transgene in these cells? Can simple IF also be used to show the cellular distribution of EndoG from these constructs – using both the EndoG antibody and HTL-TMR? This might help confirm whether the overexpression leads to subnuclear localisation that differs to endogenous EndoG.

Reply:

The reviewer is right in that we used the leaky expression of a Tet-inducible promoter to ensure low expression levels of exogenous EndoG. We now emphasize this in the methods.

As suggested, we now quantified the expression level by Western Blotting and found overexpression of Halo-tagged proteins 4.5-fold (EndoG-WT), 3.4-fold (EndoG Δ CAT), and 1.7-fold (mEndoG) compared to endogenous EndoG (see new Supplementary Fig. 10b-c). We further controlled whether these levels of overexpression affected the binding behaviour of EndoG-HT, by plotting the bound molecules versus all detected molecules of each measured cell in our single-molecule measurements. Here, the natural fluctuation of molecule numbers between cells (at constant dye concentration) enables visualizing whether the relation of bound to all molecules is linear, as expected from the law of mass action, or approaches a horizontal line, as expected for saturation of binding events. At our levels of overexpression, binding is well described by a linear relation, and not approaching a constant y-value, for all three constructs (see new Supplementary Fig. 10d).

Of note, most of the EndoG-HT is localized to mitochondria in our cells, similar to endogenous EndoG, and only few molecules enter the nucleus (Supplementary Fig. 10a).

23. There are a few issues in figure legends, e.g. “10 μ M” in the legend of Fig. 2 that should presumably read “10 μ m”.

Reply:

We carefully checked the Figure legends and amended erroneously used “ μ M” versus “ μ m”, in particular.

Reviewer #4 (Remarks to the Author):

In this manuscript, the authors describe a fragment of the Ku80 protein that is able to antagonize the EndoG enzyme and preserve the genome integrity of the MLL major breakpoint region. Starting with the observation that there is sequence similarity between an EndoG inhibitor in fruit flies and the Ku80 c-terminal tail, the team interrogates the sequence to find an optimal peptide for EndoG inhibition in mammals. Following molecular modeling, the team finds 4 fragments to test further with the one named Ku3 having the best ability to prevent instability at MLL while still allowing sensitivity to the Top2 poison doxorubicin. Overall, the idea is to use an EndoG inhibitor to allow for chemotherapy to target cancer cells but not hit fragile regions of the genome that can lead to secondary cancers. Given the significant risk for treatment-related cancers to occur with drugs that induce DNA double-strand breaks, research in this area is significant. Strengths of this work are the molecular modeling, PLA analysis, and single molecule tracking experiments. Major weaknesses are the rigor and significance of the assays monitoring DNA instability. This makes it unclear what the broader impact of this work will be regarding cancer treatment and prevention of secondary malignancy.

Reply:

We thank the reviewer for emphasizing the broader impact of our work and the expert suggestions that helped to substantially improve our work

1. Figure 1A depicts removal of the DNA-PKcs interaction domain from the Ku80 C-terminus. More recent work from Blundell (PMID: 20023628) and Lees-Miller (PMID: 28641126) indicate the importance of the 592-732 region for DNA-PKcs binding and I'm curious if the authors have been able to completely rule out interaction between Ku1-4 and DNA-PKcs.

Reply:

The two works suggested by the reviewer, although relevant for the DNA-PKcs structure and the residues involved in the interactions with Ku70/80, do not provide a tridimensional architecture of the whole complex, which is important to rule out the interactions between the peptides (specifically the lead peptide Ku3) and DNA-PKcs. For this, we employed the structure of Seif-El-Dahan et al. 2023 Sci Adv (<https://www.science.org/doi/10.1126/sciadv.adg2834>) that depicts a dimer of the DNAPKcs-Ku80-Ku70-PAXX-XRCC4-DNA ligase 4 (PDB ID: 8BH3) complex. This structure clearly shows the interaction of Ku80_Ct (residues 593-732) with DNA-PKcs. In the figure below, we highlight the Ku80_Ct domain (green ribbons).

The figure shows in red the Ku3 region (EIVVQDGITLITKEEASGSSVTAEAEAKK) in Ku80_Ct. The zoomed panels show in ribbons the DNA-PKcs areas in the surroundings of Ku80_Ct. As depicted, the Ku3 region (highlighted in red) is far away from DNA-PKcs, and does not interact with it.

2. For experiments in Figure 2 comparing Ku80-wt overexpression to Ku80-ct expression the 3 datapoints are highly inconsistent from experiment to experiment and it is unclear why everything is normalized to Ku80-wt and not the control. With such inconsistencies, more than 3 data points should have been collected. Further, Ku80 levels in most cells is already high and the connection of why Ku80 is bound to the MLL and enhancing EndoG binding is unclear to me. The effect may just be an artifact of Ku80 overexpression. Is the thinking that Ku80 is present at DNA independent of binding to Ku70? The KD of Ku70/80 binding is very low, so there would have to be a stoichiometric imbalance for Ku80 to be present alone. Can the authors overexpress Ku70 to see if the EndoG binding is truly Ku80-dependent?

Reply:

- **Ad normalization:** We normalized our ChIP values to the Ku80-wt values, because as a general rule higher reference values permit more robust normalization as compared to lower ones. Some of the ctrl values are even close to zero and therefore would not be suitable as reference values.
- **Ad data points:** Please note that very different from transcription or chromatin factors, only a small fraction of EndoG molecules enter the nucleus, as only limited introduction of chromosomal breaks is compatible with life (Gole et al. 2018 Leukemia; Liu et al. 2016 Mol Cell Oncol; new Supplementary Fig. 10a). Therefore, *MLL*bcr-specific ChIP analyses of EndoG and of γ H2AX at the *MLL*bcr, marking DNA double-strand breaks (DSBs) after local EndoG-mediated cleavage, are challenging, which explains larger standard deviations (SDs). However, we would not call such deviations “inconsistencies”, because the trend was the same in each experiment just to a different extent. To address the larger SD seen

for the ChIP- γ H2AX analysis (due to one experiment with a less pronounced effect), we newly added one more ChIP analysis and the resulting data set strengthens our original observation (Fig. 2c). Finally, please note that the ChIP values for Ku80-Ct versus Ku80-wt decrease in Figure 2c even though the input PCR using the same primers increases in Fig. 2d, indicating an underestimation rather than overestimation of the observed effect of Ku80-Ct during ChIP experiments.

- **Ad Ku80 levels:** We would like to emphasize that exogenous Ku80 was not aberrantly overexpressed according to Western blotting of total Ku80 (Fig. 1d, Fig. 2a). We can only speculate on different features of pre-existing Ku80, stabilized by Ku70 heterodimerization (Koike 2002 J Radiat Res), and de novo produced Ku80 such as regarding differential post-translational modifications that might impact on nuclear localization and/or chromatin binding. Importantly, in the revised version of the manuscript we newly show co-immunoprecipitation data of endogenous Ku80 and endogenous EndoG (Supplementary Fig. 1a). Moreover, in our previous work (Gole 2018 Leukemia) we noticed that knockdown of endogenous Ku80 diminishes replication stress treatment-induced *MLLbcr* rearrangements similarly as knockdown of EndoG (see illustration below), while I-SceI-induced homologous DSB repair of the same reporter construct was not affected by EndoG knockdown, Ku80 knockdown augmented homologous DSB repair, likely due to a pathway shift.
- **Ad Ku70:** Indeed, our reasoning was inspired by earlier screen data showing that knockdown of Ku80 and of Ku70 reduce *MLLbcr* rearrangements by 28-49%, like EndoG, while we saw a decrease only by 11% after knockdown of DNA-PKcs (Gole 2018 Leukemia; data mentioned in the revised manuscript on page 10). However, the superposition of the predicted structure of EndoG-Ku80_Ct + Ku3 from Fig. 3a onto Ku80 in the structure of DNAPKcs-Ku80-Ku70-PAXX-XRCC4-DNA ligase 4 (PDB ID: 8BH3; Seif-El-Dahan et al. 2023 Sci Adv) argues against a direct interaction of Ku70 and EndoG as well as against a direct interaction of DNA-PKcs and EndoG (see reply to Reviewer #1). Therefore, our model implies Ku complex-mediated recruitment of EndoG to specific sites in the chromatin, where Ku70, Ku80 and EndoG synergistically recognize particular secondary structures, augmenting resolution of these replication barriers by EndoG. Notably, all three proteins are known to recognize G4 and other non B-DNA secondary structures in the genome such as cruciforms or hairpins (Postow 2011 FEBS Lett; Zhang 2012 Anal Bioanal Chem; Zanden 2020 Nucleic Acids Res; Dahal 2022 Elife). We emphasize this reasoning in the Discussion section of the revised manuscript on page 17.

Chromosomal assay in HeLa reporter cells

Chromosomal assay in HeLa reporter cells (Gole 2018 Leukemia).

MLLbcr rearrangements were quantified in HeLa cells with chromosomally integrated reporter for recombination between differently mutated *EGFP* genes encompassing *MLLbcr* (see Fig. 1d). Left panel: Relative recombination frequencies with and without knockdown of EndoG (siENDOG) or Ku80 (siXRCC5) and replication stress-inducing treatment aphidicolin (10 μ M). Right panel: Relative recombination frequencies with and without knockdown of EndoG (siENDOG) or Ku80 (siXRCC5) and targeted cleavage of the reporter construct by I-SceI; * $p < 0.05$; ** $p < 0.01$; **** $p < 0.0001$.

3. Are breaks at MLL dependent on doxorubicin only or can other Top2 poisons like mitoxantrone also cause breaks there?

Reply:

As we were able to demonstrate previously, multiple genotoxic treatments causing replication stress or directly DSBs will induce loss of *MLLbcr* integrity (Ireno 2014 Arch Toxicol; Gole 2015 Oncogene; Gole 2018 Leukemia; Eberle 2021 Front Oncol).

To answer whether Ku80-Ct also affects *MLLbcr* integrity in response to topoisomerase II-poisons, we measured *MLLbcr* rearrangements after the 4 h pulse treatment with etoposide instead of doxorubicin. We do not see a significant effect of Ku80-Ct on etoposide-induced *MLLbcr* rearrangements (see new Fig. 1g), which through topoisomerase II-poisoning complex formation directly generates DSBs, while the mode-of-action of doxorubicin relies on DNA adduct formation and DNA intercalation and thereby formation of R-loops, ROS-induced replication stress and DNA-breaks (Yang 2014 BBA). There are many examples in which doxorubicin-mediated cell killing is independent of topoisomerase II. Doxorubicin was focused on here, as it has been recognized as the mainstay agent to treat advanced breast cancer despite the risk of secondary leukemia (Bondarev 2024 Pharmacol Res).

4. In experiments shown in Fig 4 combining Ku3 and doxorubicin treatment, the most the recombination frequency decreases is ~35%, would this not mean that a majority of the time breaks at MLL are still occurring to lead to rearrangement? I'm just not sure how effective Ku3 would actually be in the clinic. Also, it is puzzling that at high levels of Ku3 the protective effect is lost.

Reply:

We also noticed that downregulation of *MLLbcr* rearrangements by the Ku3 peptide was more modest than by Ku80-Ct, which caused a reduction of the recombination frequency down to 52% of the frequency measured with Ku80-wt. Given that our EndoG-Ku3 binding simulations suggest that peptide Ku3 interferes with EndoG-Ku80 complex formation and additionally modulates chromatin binding of EndoG dimers, it is tempting to speculate about a different impact of these two modes-of-action differentially affecting exo- and endonucleolytic activities of EndoG mono- and dimers, respectively. Furthermore, Ku80-Ct is predicted to interact with EndoG through multiple regions along its sequence clustering around residues 686-707 and to a lesser extent also around residues 599-624 (Fig. 4a). Therefore, in the discussion section on page 19 we propose that combining peptides may optimize binding affinities and specificities.

5. On page 6 line 143, the authors say the experiments with the Ku80 E720A and E721A mutations are inconclusive due to “data scattering”. Is this not just the consequence of the messy nature of FACS analysis that requires further data points? This explanation is used in other part so the manuscript as well. In Fig S4C and S4D, data is concluded with only an n=2, which could impact rigor and reproducibility.

Reply:

As suggested by reviewer #3 we excluded data obtained for mutant Ku80(E720A, E721A) from the manuscript. Regarding the number of experiments of n=2 in Fig. S4c and S4d, we would like to point out that in Fig. S4 we are looking at titration analyses with measurements performed at ≥ 3 increasing peptide concentrations, starting from the Ku3 peptide concentration showing a

significant effect on *MLLbcr* rearrangements. All recombination measurements in figures other than peptide titrations rely on at least three independent experiments.

Regarding the nature of FACS analysis, please note that we used similar gating strategies and detected similar numbers of EGFP+ cells as seen by other experts (see e.g. Pierce and Jasin 1999 Genes Dev; Bennardo et al. 2008 Plos Genet). Importantly, the sensitivity of these assays is determined by the total number of living cells analyzed by FACS. For chemotherapeutic treatment-induced recombination at the chromosomally integrated *MLLbcr* reporter we typically analyze 0.5-1 million viable cells to compensate for the low recombination frequency of on average 0.01%. However, for the I-SceI induced, i.e. maximally efficient DSB repair, typically 10 000-50 000 viable cells are analyzed by FACS (Pierce and Jasin 1999 Genes Dev; Bennardo et al. 2008 Plos Genet; this work), ultimately resulting in similar counts of EGFP+ cells.

6. The gel assay to monitor MLL stability seems rather low-definition. Are results from the recombination assay, which would only happen if a DSB occurred, not happening at a high enough frequency to detect differences? A quantitative digital PCR assay or even amplicon sequencing of the MLL region may provide a better readout of MLL instability.

Reply:

- Ad DSBs: Please note that homologous recombination can be triggered by multiple DNA lesions and structural changes in the DNA other than DSBs including but not limited to stalled replication forks (see e.g. Berti 2020 Nat Rev Mol Cell Biol).
- Ad quantitative PCR: We unfortunately had to apply conventional quantification of band intensities after PCR amplifications because of the proneness of the *MLLbcr* to form extended secondary structures (exacerbated by doxorubicin treatment, Yang 2014 BBA), which are known to cause problems especially during qPCR (see e.g. Fan 2019 J Biomol Struct Dyn 2019). We encountered these problems during method establishment, when also exploring the possibility of using qPCR, regardless of primer target sites, additives and other optimization measures. However, PCR band intensities were only quantified in the linear range according to our ChemiDoc™ MP Imaging System (Bio-Rad Laboratories, Hercules, California, USA). Moreover, we routinely included 50% template controls in our PCR runs (now shown in the revised manuscript in Fig. 5a and Supplementary Fig. 5c).
- Ad amplicon sequencing: Given the length of the *MLLbcr* and the rarity of recombination events (see reply to comment 5.) as compared to the depth of coverage during amplicon sequencing, it remains an open question whether such an approach will provide the necessary sensitivity to detect *MLLbcr* rearrangements in short-term culture.

7. Has previous work looked genome-wide for regions of Ku80-EndoG binding? This could be an opportunity to determine if other sites in the genome are sensitive to EndoG and if GC content, non-B structures, or DNA methylation is playing a greater role in cleavage.

Reply:

Thank you very much for this thoughtful suggestion. We already attempted to map binding of EndoG genome-wide by Cut&Run together with experts in this technology, which was not successful despite positive and negative controls working. Please note that very different from transcription or chromatin factors, only a small fraction of EndoG molecules enter the nucleus, as only limited introduction of chromosomal breaks is compatible with life (Gole et al. 2018 Leukemia; Liu et al. 2016 Mol Cell Oncol; new Supplementary Fig. 10a). Therefore, an extremely

sensitive genome-wide detection method will be needed. We believe that in-depth optimization of genome-wide analysis of Ku80-EndoG chromatin binding will go beyond the scope of this work focusing on the discovery of an EndoG interaction partner protecting the *MLLbcr* integrity.

8. The DNA assay used is homology-based. For Figure S5, what was the recombination frequency, not just the result of the gel assay? An assay that specifically looks at repair through NHEJ would be more appropriate to determine if an inhibitor of DNA-PKcs really has no change in repair frequency in the presence of Ku3 overexpression.

Reply:

As requested we performed recombination measurements $-/+DNA-PKi$ $-/+Ku3$ (new Supplementary Fig. 5b, left panel). Here, we found significant ($p<0.05$) and trendwise ($p<0.10$) mitigation of *MLLbcr* rearrangements by Ku3 in absence and presence of DNA-PKi, respectively. However, homologous DSB repair was augmented by DNA-PKi, likely due to a pathway shift, as can be seen after expression of I-SceI (Supplementary Fig. 5b, right panel). Such pathway shift could be a confounding factor for detection of the Ku3 effect. To circumvent a pathway shift during repair, we newly performed measurements of NHEJ using a specific NHEJ reporter showing that Ku3, like Ku80-Ct, did not affect NHEJ frequencies (Supplementary Fig. 2c, 5c).

Together with earlier screening data (Gole 2018 Leukemia) and the superposition of the EndoG-Ku80_Ct + Ku3 of Fig. 3a, onto Ku80 in the structure of DNAPKcs-Ku80-Ku70-PAXX-XRCC4-DNA ligase 4 (PDB ID: 8BH3; Seif-El-Dahan et al. 2023 Sci Adv, see reply to Referee #1), these data strengthen the idea of an NHEJ- and at least partially DNA-PKcs-independent effect of Ku3 on *MLLbcr* rearrangements.

REVIEWER COMMENTS

Reviewer #1 (Remarks to the Author):

The authors have addressed all reviewer comments, and I am happy for the manuscript to be published.

Reviewer #2 (Remarks to the Author):

Reply:

We sincerely thank Reviewer #1 and #2 for their interest and constructive criticism.

Reviewer #3 (Remarks to the Author):

The authors have made a substantial effort to address the concerns raised in the previous review and have added a considerable amount of new data. Many of the previously raised issues have been well addressed well. With respect to the major comments, I am satisfied with the authors' responses regarding the use and interpretation of the MLLbcr reporter assay. The explanation for the low recombination frequencies is now clear and appropriate. I also find the discussion of the partial alleviation of MLLbcr breakage by the peptide inhibitor convincing, particularly the authors' acknowledgment that the modest effect size could potentially be improved through the use of combined peptides targeting different EndoG activities. The clarification regarding the apparent PCR bands in Supplementary Fig. 5 are appreciated. The rationale provided for the inability to use qPCR at the MLLbcr is reasonable and adequately discussed in the revised manuscript. That said, it remains somewhat unusual that conventional PCR could therefore be used; it would be helpful if the band densitometry approach were explicitly described in the Methods section, clarifying that MLLbcr band intensities are normalized to appropriate internal control bands in each experiment.

Reply:

Again, we would like to express our gratitude for the diligent evaluation and excellent suggestions made by Reviewer #3.

One possible explanation for the superiority in detecting quantitative changes by conventional PCR better than via qPCR could be that conventional PCR was quantified after separating the amplification products by agarose gel electrophoresis, thereby excluding aberrant PCR products and DNA conformers.

In the newly revised version of the manuscript we explicitly explain how conventional PCR products were quantified and normalized to internal controls in the Methods section.

Reviewer #3

All minor edits have been carefully addressed. I also appreciate that the revised manuscript now clearly states that Ku80-Ct does not affect homologous recombination or non-homologous end joining as measured using I-SceI-based reporter assays (Results, Fig. 1; Supplementary Fig. 2). The inclusion of etoposide as a control (Fig. 1g) is now used consistently to support the conclusion that Ku80-Ct does not affect canonical DSB repair, and the Discussion appropriately reiterates that the effects of Ku80-Ct are independent of both DSB repair pathway choice and DNA-PKcs activity (Discussion, pp. 16–18).

I do have several minor follow-up points related to specific responses:

Regarding Ku80 overexpression, the authors state that there is only limited overexpression of the transgenic Ku80 constructs and refer to Fig. 1e and Fig. 2a. While this is likely correct, it is difficult to fully assess from FLAG-based Western blots alone, as different exposures can strongly influence the apparent degree of overexpression. On re-examination of the source data, it is also unclear why no Ku80-wt-FLAG signal is visible in the FLAG blot shown in “Source Data, Extended Fig. 1: Uncropped Western Blots of Fig. 1e.” This raises the question of whether Ku80-Ct may in fact be overexpressed to a much greater extent than Ku80-wt. In addition, the molecular weight markers shown in the uncropped images do not appear to match those in Fig. 1e (e.g., 80/14 kDa versus 70/15 kDa). These issues do not undermine the main conclusions, but addressing them would improve clarity and confidence in the data presentation.

Reply:

In the Source data of Fig. 1e we added a long exposure of the uncropped α -Flag immunoblot, visualizing two bands, namely for Flag-tagged Ku80-Ct and to a lesser extent also Flag-tagged Ku80-wt. For Fig. 2a and the Source data of Fig. 2a we chose another Western blot experiment in order to visualize on the same α -Ku80 immunoblot bands for total Ku80, i.e. in lane “ctrl” the band for endogenous Ku80, in lane “Ku80-wt” for endogenous Ku80 and exogenous Ku80-wt and in lane “Ku80-Ct” for endogenous Ku80 and for exogenous Ku80-Ct, the latter showing a weaker band intensity. We also depict a short and long exposure time of the α -Flag (HRP) immunoblot, visualizing bands for both Flag-tagged proteins, namely Flag-tagged Ku80-Ct and to a lesser extent also Flag-tagged Ku80-wt. Our conclusion that there is only limited overexpression is based on the fact that α -Ku80 band intensities are similar in lanes “ctrl” and “Ku80-wt” as well as on the fact that the short Ku80-Ct fragment in lane “Ku80-Ct” does not show a stronger α -Ku80 band intensity than endogenous Ku80. Though Flag-tagged Ku80-wt and Ku80-Ct show different band intensities in the α -Flag (HRP) immunoblot, we hesitate to conclude that Ku80-Ct is overexpressed to a much greater extent than Ku80-wt. Quantitative comparisons are difficult, because smaller proteins are transferred to Western blots much faster.

In the revised figures we show the molecular weight markers in the images of the main Figures as shown in the uncropped Source data.

Reviewer #3

The explanation regarding doxorubicin autofluorescence complicating flow-cytometry-based assays is reasonable. However, I am confused by the apparent lack of EndoG foci induction between mock- and doxorubicin-treated cells in Fig. 2b – though I appreciate that this figure is intended to highlight the EndoG recruitment by ectopic Ku80 expression. In other figures, it is not always clear whether the data represent baseline MLLbcr breakage or doxorubicin-induced effects. For example, in Fig. 2c and 2d, if the data reflect doxorubicin treatment, it would be helpful to explicitly show or reference the induction relative to mock-treated controls. In the Discussion, the authors speculate that endogenous Ku complex recruits and synergises binding of EndoG to DNA secondary structures at the MLLbcr locus after doxorubicin treatment. Do the authors expect this to apply to other fragile loci? While the differential chromatin binding of EndoG is well-reported in this study by SMT, I wonder if it could be further validated by analysing chromatin fractions for EndoG in the presence and absence of doxorubicin. This is not essential and could reasonably be left for future work, but a brief comment acknowledging this limitation would be helpful.

Reply:

The color code used throughout the manuscript, namely white columns for water (and aphidicolin), grey columns for etoposide and blue columns for doxorubicin-treated cells, was meant to indicate the type of treatment. In this 2nd revised version of the manuscript, we have added information on the specific treatment below the graph or in a color legend within the figure plus in the legend to each figure.

Indeed, in the Results section lines 182-184 we reference our previous work showing increased *MLLbcr* binding by EndoG after doxorubicin treatment in ChIP experiments (reference #30: Eberle et al. 2021 Front Oncol). Moreover, in our 2nd revised version of the manuscript we additionally provide this information in the legend of Fig. 2c. The observation of increased *MLLbcr* binding by EndoG after doxorubicin treatment in ChIP experiments was the starting point of this work, why we focused on doxorubicin-treated cells to maximize the quality of the technically challenging and long-lasting ChIP experiments. Only in early experiments we also analyzed the water controls, see the EndoG-ChIP below, indicating augmentation of *MLLbcr* binding by EndoG after doxorubicin treatment consistent with our previously published observation (reference #30: Eberle et al. 2021 Front Oncol).

Reviewer #3

Similarly, Ku3 is described as “sufficient to mitigate chemotherapy-related destabilizing effects,” whereas the observed effects are relatively modest. I recommend moderating this wording (e.g., “moderately effective” or “partially mitigates”) to better reflect the magnitude of the observed effects.

Reply:

As suggested we replaced the word “mitigate” in the context of the Ku3 effects by “partially mitigate” or “reduce” throughout the manuscript.

Reviewer #3

Finally, while the authors provide reassurance regarding ChIP antibody validation and convincingly demonstrate EndoG specificity, it remains possible that additional low-abundance bands are present but not visible at the selected exposure. This is a minor point and does not substantially detract from the conclusions, but it is worth acknowledging as a technical limitation.

Reply:

We inserted this limitation in the legend to Fig. 2c.

Reviewer #4 (Remarks to the Author):

I feel the authors have addressed my comments and the new data added to the manuscript improves it. Given the modest reduction that Ku3 expression has in the MLLbcr reporter assay with doxorubicin in Figure 4, and that high concentration mitigates this, it would be interesting if the authors eventually determine if using multiple Ku CTD fragments has a greater impact as they suggest doing in their rebuttal. This may be more convincing in showing this could improve clinical outcomes during chemotherapy treatment. Overall, though, I am supportive of publication in the current form following the revision.

Reviewer #5 (Remarks to the Author):

Reply:

We thank both reviewers for their great interest in our work and the constructive criticism. Regarding analysis of Ku80-Ct peptides targeting two EndoG interaction sites, we learnt in previous works on dual inhibitory compounds (Malka *et al.* 2021 *Biomolecules*; Marzouk *et al.* 2024 *ACS Omega*) that compound combinations are inferior to physically linked dual inhibitors. However, testing the effectiveness of such drug conjugates involves complex chemistry and subsequent screening of the resulting dual compounds with varied linker lengths. We believe that such truly interesting work must await systematic studies in the future.